# SmartSAM: Segment Ambiguous Objects like Smart Annotators

## Abstract

Segment Anything Model (SAM) often encounters ambiguity in interactive segmentation, where insufficient user interaction leads to inaccurate segmentation of the target object. Existing approaches primarily address ambiguity through repeated human-model interactions, which are time-consuming due to the inherent latency of human responses. To reduce human efforts, we propose a novel interactive segmentation framework that leverages the model's inherent capabilities to effectively segment ambiguous objects. Our key idea is to create an annotator-like agent to interact with the model. The resulting SmartSAM method mimics intelligent human annotators, resolving ambiguity with a single click and one reference instance. The agent generates multiple prompts around the initial click to simulate diverse annotator behaviors and refines the output masks by iteratively adding click chains in uncertain regions, thereby producing a set of candidate masks. Finally, the agent selects the mask that most closely aligns with the user's intent, as indicated by the reference instance. Furthermore, we formalize the agent's behavior as a fuzzy regression problem by quantifying ambiguity using fuzzy entropy. We demonstrate that our agent yields lower entropy than traditional methods, and we establish robustness and sufficiency theorems to ensure effective, human-like decision-making within a bounded range of actions. We evaluate our approach on multiple segmentation benchmarks and demonstrate its superiority over state-of-the-art methods.

## 1 Introduction

Interactive segmentation typically rely on single-turn (Boykov and Jolly, 2001; Zhang et al., 2024b; Liu et al., 2024b) or multi-turn (Huang et al., 2023; Lee et al., 2024) human guidance to predict accurate masks for desired objects. Among them, the Segment Anything Model (SAM) (Kirillov et al., 2023; Ravi et al., 2025) and subsequent works (Huang et al., 2024c; Zhao et al., 2024) have made significant progress in high-quality segmentation and show potential in medical care (Li et al., 2025), autonomous driving (Fan et al., 2023), and remote sensing (Shan et al., 2025).

A key issue with these methods is ambiguous predictions caused by insufficient interactions, where models often misinterpret the user's intent, leading to undesired segmentation masks. As illustrated in Fig. 1, when a user clicks or gazes at the target object, the model may produce an incorrect mask of the dress due to ambiguous intent, prompting the user to provide additional interactions to clarify the intent. In large-scale annotation scenarios, repeated interactions may appear feasible but can result in significant cumulative time costs due to user refinement. Recent methods (Huang et al., 2023; 2024c) have reduced inference latency to the millisecond level, a time span typically negligible compared to human actions. This suggests that the primary time cost in the scenarios mentioned above lies outside the model, and we refer to the interactions between the user and the model as **outer interactions**. Moreover, this issue is particularly pronounced in Augmented Reality and Virtual Reality (AR/VR) scenarios (Zeng et al., 2025), where a user's gaze faces challenges in facilitating multiple interactions, thereby amplifying the significance of this problem for SAM-based methods.

Previous methods (Zhao et al., 2024; Huang et al., 2024c; Chen et al., 2022; Du et al., 2023; Ke et al., 2023) often overlook the ambiguity of a single click and focus on segmenting target instances through multiple turns of outer interactions, leading to significant time consumption. In fact, using as less amount of human interactions (*e.g.*, reference instances in Reference Segmentation methods

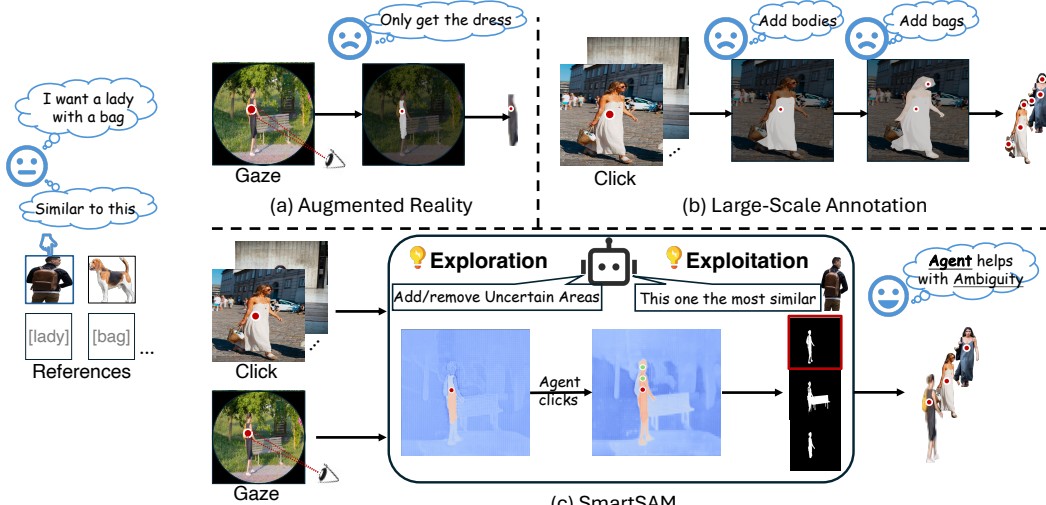

Figure 1: Ambiguity is a common obstacle for interactive segmentation methods including SAM, resulting in expensive multi-turn human interactions or imprecise single-turn human interactions. Our strategy uses an agent to overcome this by automatically interacting with itself, saving human cost.

(Zhang et al., 2024b; Liu et al., 2024b; Sun et al., 2024)) to address ambiguity can effectively improve interaction efficiency. An agent that utilizes "low response time interactions" can reduce outer interactions by automatically acting in place of humans. In contrast to outer interactions, we refer to this automatic behavior as **inner interactions**, as it occurs within the model between the agent and the segmentation model. Thus, the question arises of *how to replace the heavy outer interactions with the lighter inner interactions*. A potential solution is to have the agent behave like a human.

We observe that human interactions follow spatial patterns that can guide the agent. Specifically, the oracle outer interactions tend to fall in the middle of uncertain areas, where the model's predictions are not confident. As shown in Fig. 1, the model is confident that the dress is within the user's desired mask, while backgrounds such as the road and woods are not part of the intent. However, the predicted scores for the lady and the bags fall within the uncertain range, suggesting that the model considers them as potential instances. Therefore, we can use this pattern to build an agent and address the aforementioned question.

Therefore, we propose a training-free SmartSAM method to resolve ambiguity with a single click and one optional reference in either visual or textual form., functioning like intelligent human annotators. After clicking on the image, the agent generates diverse initial prompts based on the provided input, simulating various human annotators confronting the same image. The agent then processes each initial prompt through a series of inner interactions in the aforementioned uncertain areas to mimic human refinement, resulting in a set of candidate masks. Finally, the agent either uses the reference to select the mask that best aligns with the human intent, or just provides the mask of highest quality from the set.

Notably, Few-Shot Segmentation (FSS) (Liu et al., 2024b; Zhang et al., 2024a) and Open-Vocabulary Segmentation (OVS) (Cuttano et al., 2025) methods also utilize visual or textual references. However, these methods are inherently non-interactive and lack the capability to automatically refine predictions based on user feedback. In contrast, SmartSAM is explicitly designed to perform intelligent inner interactions while also enabling user-driven outer interactions, and can be seamlessly integrated with existing SAM-based interactive segmentation methods.

The behavior of the agent in finding and selecting can also be formalized as a fuzzy regression problem (Zadeh, 1965). We translate a model's ability to handle ambiguity into a quantitative form of fuzzy entropy. We prove that the entropy of inner interactions is always less than that of corresponding outer interactions and derive the robustness theorem for ambiguity. Furthermore, to enable the agent to effectively mimic human behavior with fewer inner interactions, we prove the sufficiency theorem of our strategy, demonstrating that the total number of the agent's inner interactions can be kept within an acceptable range.

Our contributions can be summarized as follows:

- Unlike previous work focused on outer interactions, we are the first to overcome the challenges from the perspective of inner interactions. Additionally, we propose a training-free SmartSAM method to resolve ambiguity, functioning like intelligent human annotators.
- We provide a solid theoretical analysis, including two theorems for our method from the perspective of fuzzy statistics. The first theorem focuses on the ability to deal with ambiguity, while the latter illustrates the strategy's efficiency.
- We evaluate our approach on multiple segmentation benchmarks and achieve superior performance compared to state-of-the-art methods.

## 2 RELATED WORKS

**Interactive Segmentation.** Interactive segmentation aims to segment objects in images by leveraging user interactions, such as clicks, scribbles, or bounding boxes. Traditional approaches formulate this task as an optimization problem (Adams and Bischof, 1994; Boykov and Jolly, 2001; Grady, 2006), while early deep learning-based methods integrate user interactions as auxiliary guidance channels (Xu et al., 2017; Lin et al., 2022). Subsequent research has concentrated on designing model architectures (Chen et al., 2022; Huang et al., 2023; Liu et al., 2023) to better encode user feedback. These advancements have yielded improvements across multiple dimensions, including inference efficiency (Huang et al., 2023; Du et al., 2023; Liu et al., 2024a), segmentation granularity (Zhao et al., 2024; Li et al., 2018; Liew et al., 2019), and output stability (Huang et al., 2024c; Lee et al., 2024). Despite this progress, interactive segmentation continues to face a fundamental challenge: inherent ambiguity in user intent. This often necessitates iterative refinement of user inputs to achieve satisfactory segmentation results.

**Segment Anything Model.** Recently, SAM (Kirillov et al., 2023) has advanced the field by introducing a large-scale pretrained model with promptable capabilities. SAM accommodates diverse input prompts to generate high-quality segmentation outputs. This flexibility has catalyzed several new research directions. Some studies aim to enhance performance by refining their architectural components and training strategies (Ravi et al., 2025; Zhao et al., 2024; Ke et al., 2023; Huang et al., 2024c). Others extend SAM to broader applications by incorporating multi-modal and multi-prompt interactions (Wang et al., 2024; Zhang et al., 2024c; Ye et al., 2024; Zhao et al., 2023; Li et al., 2024; Cuttano et al., 2025). A parallel line of work investigates alternative inference procedures to better exploit SAM's capabilities (Sun et al., 2024; Liu et al., 2024b; Zhang et al., 2024b). For instance, Graco (Zhao et al., 2024) introduces granularity control to allow users to adjust the precision of segmentation masks. Despite these advances, many of these approaches still depend on iterative user input for refinement. Therefore, our objective is to minimize the interaction burden ideally requiring only a single click without compromising segmentation quality.

**Agent Prompting.** Agent-based methods have garnered increasing attention in computer vision due to their flexible and interactive nature (Carion et al., 2020; Anderson et al., 2018; Park et al., 2020). These methods typically involve decision-making policies that iteratively refine predictions or explore spatial representations. Depending on the design of the decision-making policy, agents can be broadly categorized into rule-based and LLM-based paradigms. Recently, researchers have begun to explore the integration of agents with the SAM to improve its performance in complex scenarios. Several works have proposed agent-based strategies to adapt or optimize prompts during inference (Huang et al., 2024a; Ren et al., 2024; Xie et al., 2024; 2025). While LLM-based agents offer strong generalization and reasoning capabilities, they typically require high-quality trajectory data or reinforcement learning, both of which demand substantial training. Therefore, we propose a training-free rule-based agent that autonomously performs internal interactions following a single user click, thereby minimizing user involvement while maintaining segmentation accuracy.

## 3 METHODOLOGY

### 3.1 PRELIMINARIES

The Segment Anything Model (SAM) (Kirillov et al., 2023) is a foundation model for image segmentation, comprising three components: an image encoder ($E_{\text{img}}$), a prompt encoder ($E_{\text{pr}}$), and a

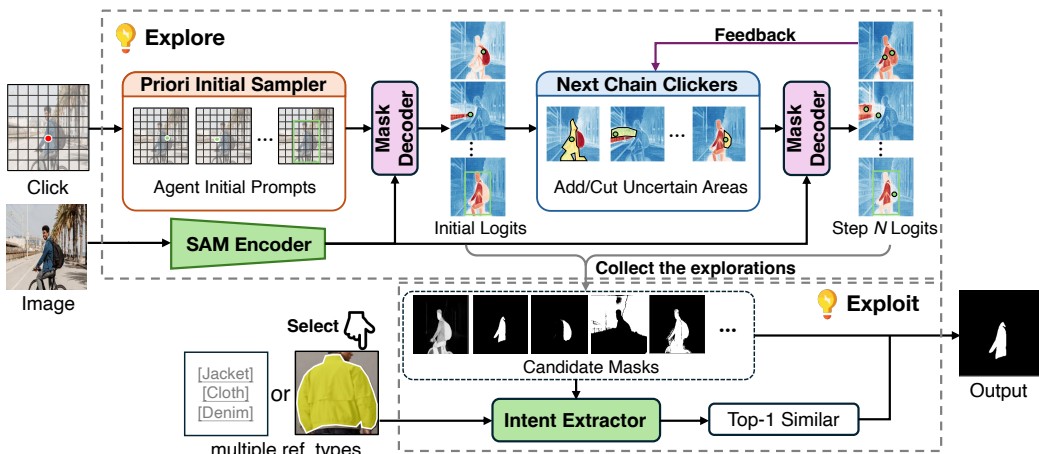

Figure 2: The framework of SmartSAM. The user clicks on the image and provides a reference instance (i.e., an image with a mask). The agent explores and generates mask candidates. The final mask is selected according to the similarity between candidates and the reference.

mask decoder ($D_{\text{mask}}$). First, $E_{\text{img}}$ extracts features from the input image, while user prompts—such as points, boxes, or masks—are processed by $E_{\text{pr}}$ to generate prompt embeddings. Then, $D_{\text{mask}}$ decodes the image and prompt embeddings to produce one or three segmentation masks, each accompanied by an estimated IoU score. In interactive settings, user prompts typically consist of positive or negative point clicks. The above process is considered a single interaction turn.

## 3.2 SMARTSAM

The agent resolves ambiguity through an explore-and-exploit strategy (see Fig. 2): it first explores a set of candidate masks, then exploits the one that best matches the user's intent. This leads to two central questions: (1) *how can the agent effectively and efficiently incorporate the user's intent during exploration*, and (2) *how can it leverage the exploration results to accurately identify the intended mask?*

**Explore the candidates.** Although the Everything Mode of SAM can explore diverse mask proposals by uniformly placing point prompts across the entire image, the process is time-consuming and does not leverage the user's click for targeted guidance. In ambiguous scenarios, SAM often fails when the user's click lands on a suboptimal region. As illustrated in Fig. 3, SAM returns a full-body mask even when the user intends to segment only the jacket. However, placing the click on a more appropriate nearby region can yield the correct mask. Motivated by this observation, we introduce fewer but more proper prompts to balance efficiency and segmentation accuracy.

Specifically, the geometric centers of the agent's prompts satisfy the following criteria: Let $\vec{w} :=$ $(x_1 - x_0, y_1 - y_0)$ denote the displacement vector between the user's input point $(x_0, y_0)$ and the agent-sampled point $(x_1, y_1)$, satisfying:

$$\|\vec{w}\|_2 \sim \beta \cdot \Gamma(2,1), \qquad \theta \sim \text{Unif}(0, 2\pi) \tag{1}$$

where $\beta > 0$ is a scaling factor, $\theta$ is the angle satisfying $\vec{w} = \|\vec{w}\|_2 \cdot (\cos\theta, \sin\theta)$, and $\Gamma$, Unif are the Gamma and Uniform Distributions. In addition, human annotators often use box prompts during annotation. Thus, motivated by early anchor-based methods (Redmon et al., 2016), which generate multiple boxes of varying shapes (aspect ratio: 0.67, 1 or 1.5), and sizes (longest side length: 200-800 pixels), we also randomly sample box prompts to explore a broader set of candidate masks, as illustrated in Fig. 3.

Moreover, the Everything Mode does not support follow-up operations to add or remove unintended regions for refinement. Consequently, the agent lacks the ability to determine where and how to refine the mask without user guidance. Fortunately, as previously discussed, most rational human segmentation behaviors can be broadly categorized as follows:

- When the mask overshoots, users typically refine it using negative clicks. Accordingly, we define Action Cut: placing negative points on uncertain regions inside the current mask.

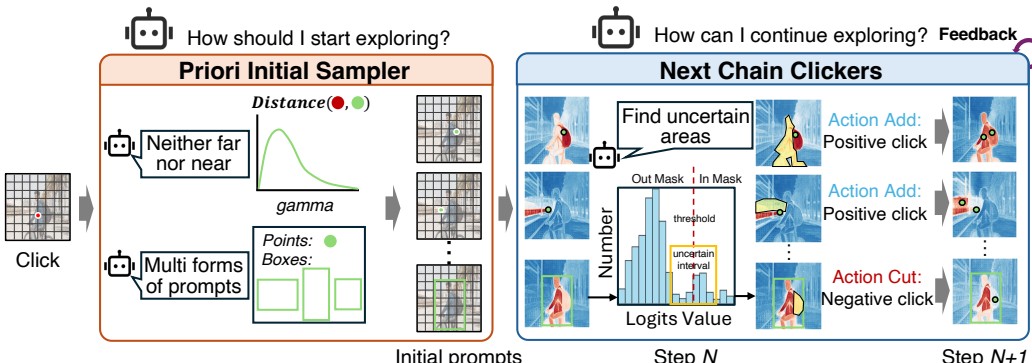

Figure 3: The framework of our agent strategy. First, the agent samples several initial prompts. Then, the agent clicks positive points on the uncertain areas outside the mask to add-on and negative points inside the mask to cut-off for refinement.

- When the mask under-segments, users typically expand it using positive clicks. Accordingly, we define Action Add: placing positive points on uncertain regions outside the current mask.

We use the scores of the IoU predicted head in SAM as the standard of evaluating whether current mask is "overshooting" or not. When the score of a mask is relatively low across the current candidates, it's more likely to need an extra positive click to add on the potential missing parts. Otherwise, when the score is high, both Cut and Add are rational. Thus, the Next Chain Clickers (NCC) decides to click positive clicks (0) or negative clicks (1) based on distribution $Bernoulli(h_{iou})$, where $h_{iou}$ is the IoU score.

These actions are organized into multiple parallel search chains, and a set of candidate masks $\mathcal{M} = \{m_i\}$ is obtained by iteratively applying the agent for refinement—mimicking the iterative behavior of human annotators.

**Exploit the candidates.** SAM can generate multiple candidate masks (i.e., multi-mask mode) but requires annotators to manually scan and select the appropriate one—introducing additional manual effort. To alleviate this burden, researchers (Liu et al., 2024b) commonly employ off-the-shelf pre-trained semantic encoders (Oquab et al., 2024; Radford et al., 2021) as the intent selector ($F_{\text{sel}}$) to infer the user's intent. Specifically, a reference feature is pre-extracted by $F_{\text{sel}}$, and the mask with the highest cosine similarity is selected. To formalize:

$$m_{\text{out}} = \arg \max_{m_i \in \mathcal{M}} \text{Cos-Sim}\left[F_{\text{sel}}(\text{img}, m_i), F_{\text{sel}}(\text{ref})\right] \tag{2}$$

When $F_{\text{sel}}$ receives an image–mask pair as input, it outputs the masked average-pooled semantic feature. When provided with only a text caption or an image, it instead outputs the corresponding class token.

### 3.3 THEORETICAL ANALYSIS

From a fuzzy-entropy perspective, we show our method surpasses the backbone because the fuzzified candidate set exhibits lower uncertainty. We first recall key notions in fuzzy statistics, then cast the ambiguity in interactive segmentation as a fuzzification problem (Zadeh, 1965). This yields a one-dimensional Fuzzy Number representation with tractable analysis; leveraging fuzzy-entropy tools, we establish a quantitative performance-improvement theorem. Finally, we recast the agent's radial search as a graph problem and derive an Efficiency Theorem.

**Definition 3.1** (Fuzzy Set and Regression (Zadeh, 1965)). *If $\Omega$ is some set, then a fuzzy subset $\overline{A}$ of $\Omega$ is defined by its membership function, written $\overline{A}(x)$, which produces values in $[0, 1]$ for all $x \in \Omega$, and thus in our work representing the probability of $x$ belongs to $A$. Similar to classic regression tasks, the target is also to optimize a model $f_\theta$ that projects $x$ to $y$ from a distribution $P(X, Y)$ based on a set of sampled data $(x_i, y_i)$. The main difference is that the regressed $y_i$ is no longer a fixed number but a variable with randomness.*

A critical fuzzification bridges the set of masks into a fuzzy set can be formalized as follows.

**Assumption 3.2** (Discrete Fuzzifization Assumption). *Human interaction can turn a set of candidate masks into a fuzzy set based on human preference. Besides, for SAM-based methods, we acknowledge an experimental observation (see Appendix for details): the set of potential masks $A$ is a subset of $\{\cup_{i_1,i_2,...,i_k}\{Area_{i_j}\}_{j=1}^k | Area_{i_j} \in \mathcal{L}\}$, where $\mathcal{L}$ is the Logits areas divided by different thresholds and gapping edges.*

Based on this, the ambiguity problem can be transformed as:

**Definition 3.3** (Problem Setting). *For interactive segmentation methods including SAM with points prompts $\{p_i = (crd_{h_i}, crd_{w_i})\}$, the all potential masks that contains $\{p_i\}$ is denoted as crisp subset $A(\{p_i\})$. The goal of disambiguity is to fuzzify $A$ into $\overline{A}(\{p_i\})$ with uncertainty.*

Note that the uncertainty of fuzzy set is evaluated by the fuzzy entropy, we give

**Lemma 3.4** (Entropy Metric for Subjective Uncertainty (Wang and Chiu, 1999)). *let*

$$h(u) \triangleq \begin{cases} 2u & \text{if } u \in [0, \frac{1}{2}], \\ 2(1-u) & \text{if } u \in [\frac{1}{2}, 1], \end{cases}$$

*then for triangular fuzzy number, the global entropy $H(\overline{A}) := \int_{x \in X} h(\overline{A}(x))p(x)\,dx \propto |\text{Supp}(\overline{A})|$, where $|\text{Supp}|$ is the support size of $\overline{A}$.*

With the lemma, we prove a theorem that demonstrates its quantitative performance gains compared to the backbone.

**Theorem 3.5** (Theorem of Inner-Interaction Robustness on Ambiguity). *Let $P_M$ be the distribution of human favor and $M_0$ the sampled candidate mask. We can project the mask space into real number space through $f_d(M_i) := \frac{IoU(M_i, M_0)}{e^{N_0 - N_i}}$, where $N_i$ is the number of points-prompt for SAM to get $M_i$. Then $\overline{A}(x; 0, 1, e^{N_{max}})$ is a triangular fuzzy number. Similarly, the related concepts can be extended to the backbone SAM and lead to a fuzzy number $\overline{B}$. Moreover, the fuzzy entropy $\int H(\overline{A})d(P_M) \leq \int H(\overline{B})d(P_M)$.*

However, in interactive scenarios, inference time is also an important consideration. The following theorem demonstrates that our method balances both efficiency and performance.

**Theorem 3.6** (Efficiency Theorem). *With $\sqrt{N_{max}}$ branches and $\sqrt[4]{N_{max}}$ iters per branch, our strategy can search out the full set $A_{all}$.*

## 4 EXPERIMENTS

### 4.1 EXPERIMENTAL SETUP

**Datasets.** Following prior work, we evaluate our method on the DAVIS (Perazzi et al., 2016) and PartImageNet (He et al., 2022) datasets. In addition, we construct a novel dataset to explicitly address ambiguity issues, as highlighted in SAM (Kirillov et al., 2023). Amb-Occ, targets occlusion-based ambiguity by selecting COCO (Lin et al., 2014) categories that are occluded by small objects in the LVIS (Gupta et al., 2019) dataset. We filter out densely clustered instances of the same category to remain within our research scope.

**Evaluation Metrics.** *(1) 1st click IoU (mIoU@1):* mIoU@1 refers to the IoU (Intersection over Union) after the first click. In our ambiguity-aware design, we primarily focus on the IoU of the first click. mIoU@1 holds the most significant importance in our evaluation because as the number of clicks grows, user input no longer exhibits ambiguity. *(2) Ratio of masks greater than IoU k (Ratio@K):* In addition, we introduce a complementary metric as the proportion of samples where the desired IoU is achieved with only one user click. This metric provides a direct measure of our method's effectiveness in low-interaction scenarios. *(3) Number of Click (NoC):* NoC refers to the number of clicks required in interactive segmentation to achieve a specified IoU. We adopt this evaluation metric to stay in line with previous methods (Huang et al., 2024c; Chen et al., 2022; Huang et al., 2023). For example, NoC@75 indicates the average number of clicks needed to achieve an IoU of 75%.

**Implementations.** Our agent first initializes 9 prompts containing 6 points and 3 boxes. For every prompt, the agent will do a sequence of 3 following actions. We adopt DINOv2-B (without register

Table 1: Comparison experiments with SOTA interactive segmentation models. We report the 1st Click IoU (mIoU@1) and the ratio of masks meeting the given IoU threshold (Ratio@75). Results show that our strategy can effectively enhance SAMs' ability to resolve ambiguity.

| Methods | Backbone | DAVIS | | PartImageNet | | Amb-Occ | |
|---|---|---|---|---|---|---|---|
| | | mIoU@1 | Ratio@75 | mIoU@1 | Ratio@75 | mIoU@1 | Ratio@75 |
| FocalClick (Chen et al., 2022)$_{CVPR_{22}}$ | SegFB3-S2 | 71.07 | - | - | - | - | - |
| InterFormer (Huang et al., 2023)$_{ICCV_{23}}$ | - | 76.84 | - | - | - | - | - |
| SimpleClick (Liu et al., 2023)$_{ICCV_{23}}$ | ViT-H | 72.50 | - | - | - | - | - |
| HQ-SAM (Ke et al., 2023)$_{NeurIPS_{23}}$ | ViT-B | 39.38 | 25.51 | 32.85 | 21.18 | 34.40 | 14.69 |
| +SmartSAM | ViT-B | 57.86 | 37.39 | 59.06 | 43.19 | 42.18 | 22.52 |
| HQ-SAM (Ke et al., 2023)$_{NeurIPS_{23}}$ | ViT-H | 45.82 | 30.83 | 45.53 | 37.41 | 41.02 | 23.14 |
| +SmartSAM | ViT-H | 59.35 | 36.65 | 65.01 | 54.69 | 48.12 | 30.54 |
| SAM (Kirillov et al., 2023)$_{ICCV_{23}}$ | ViT-B | 39.53 | 25.80 | 33.35 | 21.47 | 34.35 | 15.12 |
| +SmartSAM | ViT-B | 58.75 | 35.94 | 59.23 | 43.85 | 42.35 | 22.01 |
| SAM (Kirillov et al., 2023)$_{ICCV_{23}}$ | ViT-H | 45.97 | 31.59 | 45.53 | 37.25 | 37.81 | 22.67 |
| +SmartSAM | ViT-H | 58.57 | 40.13 | 64.53 | 53.57 | 47.19 | 30.54 |
| FocSAM (Huang et al., 2024c)$_{CVPR_{24}}$ | ViT-H | 74.62 | 64.35 | 28.47 | 20.31 | 37.98 | 24.16 |
| +SmartSAM | ViT-H | 78.32 | 74.20 | 63.60 | 51.24 | 43.78 | 25.07 |
| HRSAM (Huang et al., 2024b)$_{Arxiv_{24}}$ | ViT-H | 79.19 | 71.30 | 65.93 | 55.77 | 38.59 | 16.07 |
| +SmartSAM | ViT-H | 80.72 | 74.20 | 70.33 | 56.65 | 41.24 | 19.93 |
| SAM2.1 (Ravi et al., 2025)$_{ICLR_{25}}$ | ViT-B+ | 62.25 | 52.17 | 51.57 | 49.04 | 44.57 | 30.17 |
| +SmartSAM | ViT-B+ | 76.80 | 69.27 | 77.31 | 72.21 | 51.85 | 36.15 |

Table 2: Comparison on 1st Click IoU for SOTA FSS/OVS. SmartSAM outperforms on both datasets.

| Method | Intend Selector | Backbone | Davis | Amb-Occ |
|---|---|---|---|---|
| **FSS** | | | | |
| PerSAM | – | ViT-H | 53.77 | 27.64 |
| Matcher | DINOv2-L | ViT-H | 46.41 | 46.75 |
| GF-SAM | DINOv2-L | ViT-H | 68.21 | 39.19 |
| **OVS** | | | | |
| SAMWISE | RoBERTa | ViT-L | 44.35 | 37.04 |
| SAM | – | ViT-H | 45.97 | 37.81 |
| +SmartSAM | DINOv2-B | ViT-H | 58.57 | **47.19** |
| +SmartSAM | DINOv2-L | ViT-H | **70.74** | 46.64 |
| +SmartSAM | CLIP-B | ViT-L | 50.62 | 42.41 |

Table 3: Multi-mask evaluation (ViT-H). We report the Best IoU (denoted as $IoU_{best}$, higher is better) and NoC@90 (denoted as $NoC_{90}$, lower is better).

| Baseline | DAVIS | | PartImageNet | | Amb-Occ | |
|---|---|---|---|---|---|---|
| | $IoU_{best}$ ↑ | $NoC_{90}$ ↓ | $IoU_{best}$ ↑ | $NoC_{90}$ ↓ | $IoU_{best}$ ↑ | $NoC_{90}$ ↓ |
| SAM | 42.97 | 5.73 | 50.92 | 6.14 | 37.81 | 12.73 |
| +SmartSAM | 85.38 | 5.56 | 84.08 | 5.48 | 74.92 | 10.62 |
| FocSAM | 74.62 | 5.29 | 28.64 | 4.97 | 28.07 | 7.93 |
| +SmartSAM | 84.14 | 5.24 | 83.22 | 4.51 | 73.19 | 7.31 |
| HQSAM | 45.82 | 5.10 | 41.02 | 5.64 | 45.53 | 12.14 |
| +SmartSAM | 87.42 | 4.86 | 86.30 | 5.40 | 75.01 | 10.30 |

tokens) as the semantic encoder. All experiments are conducted on a single NVIDIA RTX 4090 GPU. For reference image preparation, we apply background removal and cropping to ensure the reference occupies approximately 70% of the original image size. When compared with FSS and OVS methods, the inputs are controlled the same since we additionally add the user clicks as the point supervision.

## 4.2 MAIN RESULTS

To evaluate the effectiveness of our strategy in addressing ambiguity, mIoU@1, Ratio@75, Ratio@85, and NoC. For DAVIS and PartImageNet datasets, we follow the click-simulating settings in (Huang et al., 2024c; Zhao et al., 2024) where the 1st clicks are placed in the middle of the ground truth masks (see the left picture in Fig. 19). To simulate the ambiguous situation of Amb-Occ, we randomly select one of the smaller occluding objects' ground truth as the 1st click area and regarding the whole object as the target ground truth.. All reported results are averaged over five independent trials to ensure statistical robustness.

Our method consistently outperforms existing approaches across all evaluated metrics. In particular, we observe a substantial improvement in mIoU@1 (see Tab. 1), which underscores the strength of our ambiguity-aware design in producing accurate masks from minimal user input. Moreover, SmartSAM consistently surpasses the FSS and OVS methods (see Tab. 2). Interestingly, some methods such as FocSAM underperform their own backbone models in the early stages of interaction. We attribute this to a trade-off between segmentation stability and local adaptability: these methods tend to over-focus on local refinement, which limits generalization when user input is sparse. Comprehensive visual comparisons and case studies are provided in the supplementary material.

Table 4: Time Cost of encoder on 4090 GPU with Batch Size 2. Results show that the additional encoding time introduced by the intent selector is negligible, and the peak memory usage remains within an acceptable range.

| Method | Backbone | DINO | Time (s) | VRAM$_{pk}$ (MB) |
|---|---|---|---|---|
| Matcher | ViT-H | ViT-L | 1.36 | 7476.30 |
| SAM | ViT-H | – | 0.84 | 4490.70 |
| +SmartSAM[*] | ViT-H | ViT-B | 0.85 | 5998.78 |
| +SmartSAM[#] | ViT-H | ViT-B | 0.92 | 5328.98 |

[*] Parallel: forward the SAM encoder and the intent selector.
[#] Sequential: forward the SAM encoder, then the intent selector.

Table 5: Time Cost of Decoder on 4090 GPU with Batchsize 2. The number of initial sampling is referred to as **# I.S.** for simplicity.

| Method | # I.S. | Time (s) |
|---|---|---|
| SAM | 1 | 0.0670 |
| +SmartSAM | 2 | 0.0711 |
| +SmartSAM | 6 | 0.0782 |
| +SmartSAM | 9 | 0.1395 |

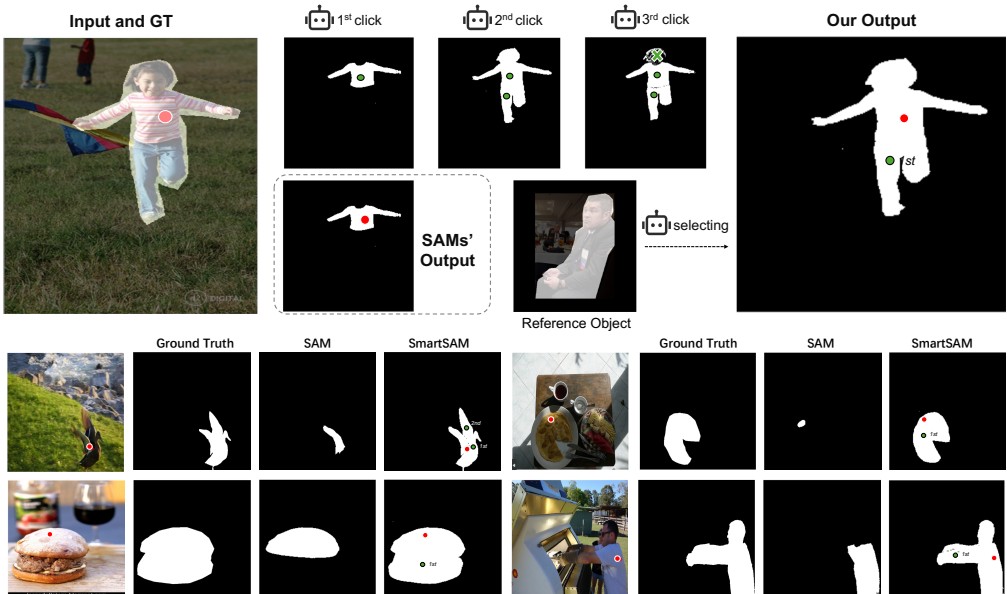

Figure 4: An example for the working flow of the agent. With one click ●, SAM only outputs part of the target. However, the agent can continue to add clicks ● and find the best matching mask.

### 4.3 EFFICIENCY STUDIES

**Agent prompts efficiency.** Given that the original SAM framework produces three candidate masks per interaction, allowing users to manually select the most suitable one as a means to address segmentation ambiguity, we also evaluate this human-in-the-loop selection paradigm for comparison. To ensure a fair assessment, we adopt the IoU of the top-1 candidate mask after the first user click as our evaluation metric, reflecting a realistic usage scenario. The results in Tab. 3 indicate that our method consistently surpasses the SAM candidate selection approach in terms of the NoC@90 and top-1 IoU across all datasets.

**Time and computation cost efficiency.** As illustrated in Fig. 2, we only perform a single forward pass through the encoder. The main additional time and computation come from the intent selector and multiple calls to the mask decoder. Since the intent selector is executed in parallel with the encoder, and we utilize batch inference of SAM to generate masks, the extra inference time required to process both the target and reference images using DINO is less than 10% of the time taken by SAM inference (see Tab. 5). Furthermore, since we adopt DINOv2-B as the intent selector, the additional time compared to the baseline SAM is controlled within 2% (see Tab. 4). Compared to the FSS method Matcher, our approach also achieves a 25% reduction in peak VRAM usage (see Tab. 4).

### 4.4 METHOD ANALYSIS

**Agent Working Mechanism.** The agent is designed according to the statistical priors of the dataset's initial distribution provided by SAM (Fig. 5 in (Kirillov et al., 2023)). We evaluate the effectiveness of our inner-interaction strategy both qualitatively and quantitatively. Qualitatively, we present failure cases of using a single initial prompt and illustrate how our approach escapes these

Table 6: Ablation studies of agent actions on mIoU@1 metric. Priori Initial Sampler is denoted as PIS and Next Chain Clicker is denoted as NCC.

| PIS | NCC | Amb-Occ | PartImageNet |
|---|---|---|---|
| Random | - | 49.79 | 72.90 |
| ours | - | 72.91 | 83.17 |
| - | Random | 49.91 | 75.31 |
| - | ours | 70.49 | 79.04 |
| - | - | 37.81 | 50.92 |
| ours | ours | 74.92 | 84.08 |

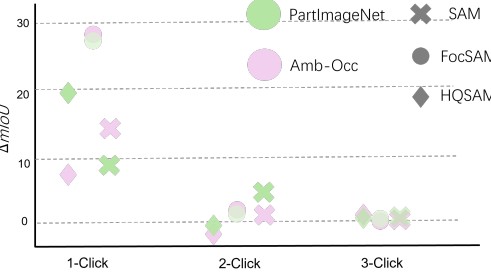

Figure 5: mIoU gains on different datasets with SAM, FocSAM and HQSAM backbones.

"traps." Quantitatively, we measure performance improvements in terms of mIoU@1, comparing our initialization to standard alternatives.

As illustrated in Fig. 4, the logits generated from a naive user click tend to activate only a partial region of the ground truth, often failing to distinguish the true target from surrounding distractors. In contrast, our method initiates a more effective agent-based sampling process, which successfully escapes these local minima. Furthermore, the quantitative results in Tab. 7 confirm that our approach consistently achieves superior performance. These findings also indicate that such situations are common in ambiguous scenarios.

**Agent Action Analysis.** We design different agents to handle various scenarios. To evaluate their contributions, we analyze the number of inner interactions required to achieve correct predictions. As illustrated in Tab 7, Action Add generates most of the predictions. Combined with Fig. 4, this implies that SAM exhibits a strong bias toward segmenting a small object encompassing the user's click location.

Table 7: Analysis on diverse actions. We count images whose predicted mask comes from different actions. Action 1 contributes the most, while other actions are necessary for harder cases.

| Inner Interaction | PartImageNet | | | | | Amb-Occ | | | | |
|---|---|---|---|---|---|---|---|---|---|---|
| Number Type | 1 | 2 | 3 | 4 | All | 1 | 2 | 3 | 4 | All |
| ⊞ Add | 1089 | 888 | 93 | 69 | 2139 | 944 | 593 | 134 | 83 | 1754 |
| ⊟ Cut | – | 48 | 34 | 3 | 85 | – | 44 | 42 | 22 | 108 |
| ⊡ Add&Cut | – | – | 115 | 139 | 254 | – | – | 126 | 219 | 345 |
| No Action | – | – | – | – | 39 | – | – | – | – | 23 |
| Ours | 1089 | 1026 | 342 | 211 | **2705** | 946 | 637 | 302 | 312 | **2385** |

Consequently, Action Add can expand the search space step by step, leading to a more accurate mask. Meanwhile, we observe that most predictions are made within the first two inner interactions, with diminishing returns in subsequent interactions. This aligns with our theoretical expectation that ambiguity is typically resolved within the first few clicks.

## 4.5 ABLATION EXPERIMENTS

As supported by our theoretical analysis, the proposed strategy is designed to mitigate ambiguity in interactive segmentation. To validate its generality and effectiveness, we conduct ablation studies by integrating our approach with several baseline methods. The results presented in Tab. 6 show that our method consistently improves performance over the baseline models. Notably, according to Fig. 5, the performance gains diminish as the number of clicks increases, approaching zero after 2-3 interactions. This trend aligns with our theoretical expectation: ambiguity tends to be resolved within the first few clicks, making further improvement less pronounced.

In particular, our approach yields substantial improvements when combined with backbones designed to enhance the stability of SAM. Such models typically constrain logits to local regions, which can lead to performance degradation in ambiguous single-click scenarios. Our strategy enables these models to overcome local traps and expand the search space, thereby significantly boosting their robustness in early interactions.

Table 8: FSS, PS, and IS methods on FSS benchmarks (COCO-20i, PASCAL-5i) and IS benchmarks (DAVIS, Amb-Occ). SmartSAM consistently improves SAM on both FSS-style and IS settings, while specialized FSS methods remain strongest on FSS benchmarks.

| Method | Task Scope | Reference | COCO-20i$^{FSS}$ $*$ | PASCAL-5i$^{FSS}$ $\dagger$ | DAVIS$^{IS}$ | Amb-Occ$^{IS}$ |
|---|---|---|---|---|---|---|
| ProSAM | FSS | yes | 48.74 | 75.26 | - | - |
| VRP-SAM | FSS | yes | 48.10 | 73.90 | - | - |
| Matcher | FSS | yes | 47.61 | 72.92 | 46.41 | 46.75 |
| PerSAM | PS | yes | - | - | 53.77 | 27.64 |
| FocSAM | IS | no | 36.22 | 41.58 | 74.62 | 37.98 |
| w/ SmartSAM | IS | optional$^{**}$ | 39.04 | 45.63 | 78.32 | 43.78 |
| SAM | IS | no | 34.71 | 40.27 | 45.97 | 37.81 |
| w/ SmartSAM | IS | optional$^{**}$ | 38.93 | 45.79 | 58.57 | 47.19 |

$*$ Results on fold-0 of COCO-20i.
$\dagger$ Results on fold-0 of PASCAL-5i.
$^{**}$SmartSAM has a pure IS mode; here we report the variant with references.

Table 9: SmartSAM in pure IS and reference modes on IS benchmarks. Pure IS mode of SmartSAM that work without references still show superiority to the baseline SAM.

| Method | Mode | DAVIS | Amb-Occ |
|---|---|---|---|
| SAM | pure IS | 45.97 | 34.35 |
| SmartSAM | pure IS$\dagger$ | 59.71 | 43.94 |
| SmartSAM | textual | 44.35 | 37.04 |
| SmartSAM | visual(DINOv2-B) | 58.57 | **47.19** |
| SmartSAM | visual(DINOv2-L) | **70.74** | 46.64 |

$\dagger$ Select the mask from the candidate set that: 1. contains the user's 1st click; 2. of the highest IoU score predicted by SAMs' IoU predict head.

Table 10: Robustness of SAM and SAM2 to TETRIS-style Moskalenko et al. (2024) first-click perturbations, with and without SmartSAM. SmartSAM consistently improves robustness under attacked first clicks and also boosts clean performance.

| Method | DAVIS | DAVIS (attack) |
|---|---|---|
| SAM | 39.53 | 33.27 |
| **w/ SmartSAM** | **58.75** | **58.52** |
| SAM2 | 62.25 | 59.66 |
| **SAM2 + SmartSAM** | **76.80** | **75.93** |

All numbers are IoU scores (%). "attack" denotes TETRIS-style first-click perturbations.

### 4.6 Extented Experimental Analysis.

**Why FSS metrics are not suitable for IS evaluation.** Table 8 compares FSS, PS, and IS methods on both FSS and IS benchmarks: specialized FSS methods (ProSAM, VRP-SAM, Matcher) perform best on COCO-20i and PASCAL-5i, while SmartSAM mainly improves SAM and FocSAM on DAVIS and Amb-Occ. This indicates that FSS-style metrics are not well aligned with click-based IS behavior and that SmartSAM should primarily be evaluated under IS protocols.

**Pure IS mode of SmartSAM still works.** Table 9 disentangles the effect of references by comparing SmartSAM in pure IS mode (no text or visual exemplars) and in reference-based modes. Even without any reference, SmartSAM substantially outperforms SAM on DAVIS and Amb-Occ, while textual/visual references with a stronger encoder (DINOv2-L) bring additional gains rather than being the sole source of improvement.

**SmartSAM is robust for real-world annotators.** Table 10 studies robustness to non-ideal first clicks using a TETRIS-style Moskalenko et al. (2024) to simulate real-worl first clicks. SmartSAM consistently boosts SAM and SAM2 in both clean and "attack" settings, with larger relative gains under perturbed first clicks, indicating increased tolerance to imperfect user clicks.

## 5 Conclusion

SAM provides a powerful backbone for interactive segmentation. However, its stability in real-world applications is often compromised, particularly in ambiguous scenarios. This limitation arises from overly fuzzy inputs, where SAM struggles to generate sufficient candidate masks and output the correct mask based on user preferences. To address these challenges, we propose a training-free SmartSAM method. SmartSAM leverages multiple chains of agents to automatically introduce points at appropriate areas, constructing a comprehensive pool of candidate masks. The most matched mask is then selected through a feature similarity comparison process. As a result, SmartSAM not only achieves state-of-the-art segmentation quality but also demonstrates remarkable performance in handling ambiguous scenarios. These advancements underscore SmartSAM's potential for broader and convenient real-world applications.

## ETHICS STATEMENT

All authors have read and agree to abide by the ICLR Code of Ethics. This work does not involve interventions with human participants or personally identifiable information. We use only publicly available datasets under their original licenses and follow the terms of use. Potential risks and our mitigations are summarized below:

- **Privacy & Security.** We do not collect or release any personal data. When showing qualitative examples, all images/videos are from public datasets; any sensitive content is filtered.
- **Bias & Fairness.** We report results on multiple benchmarks and provide detailed settings to facilitate external auditing. We acknowledge possible dataset biases and encourage follow-up evaluation on broader demographics and domains.
- **Dual Use / Misuse.** The method could be misused to enable undesired large-scale labeling or surveillance. To reduce misuse, we release only research artifacts (code/configs) and exclude any tools for scraping or re-identifying individuals.
- **Legal Compliance.** We comply with licenses of all third-party assets (code, models, and datasets) and cite their sources. Any additional third-party terms are respected.
- **Research Integrity.** We document preprocessing, training recipes, and evaluation protocols; random seeds and hyperparameters are provided to enable reproducibility.

Where applicable, institutional review information is withheld for double-blind review and can be provided after acceptance.

## REPRODUCIBILITY STATEMENT

We include training and evaluation details in the main paper and Appendix. Concretely: (i) all hyperparameters, optimization settings, and compute budgets; (ii) full data preprocessing and splits; (iii) code structure with scripts to reproduce the main tables and figures; (iv) checkpoints and logs for the primary models will be open-sourced upon paper acceptance.

For theoretical results, we provided the proofs and assumptions in Appendix.

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

# A PROOFS FOR THE THEOREMS

## A.1 THEOREM 1

*proof of Theorem 1.* First, we prove that the defined projection is a fuzzy number by indicating that it is convex and only attains the maximum value at one point. Since $e^{-x}$ is convex and our transform inside is a linear projection, the convexity can be ensured. Moreover, since $N_i \geq N_0$, the maximum is only reached when $N_i = N_0$. Also, $\text{IoU}(M_i, M_0)$ is only achieved when $M_i = M_0$. Thus, the projection has only 1 maximum point.

Since our strategy contains the output of the baseline model, $B \in A$. We denote the favor mask in $A$ as $M_0$ and in $B$ as $m_0$. Thus, by the extension principle

$$h(\overline{A}) = \int h(\overline{A}(x)) \, dP(x) = \sum_{i=1}^{|A|} p_{\alpha_i} \cdot \int h(A_{\alpha_i}(x)) dP(x) \tag{3}$$

where $|A|$ denote the cardinality (a.k.a. number of the members) of $A$ and $A_{\alpha_i}$ denote the hierarchy $\alpha$-cut based on projection $f_d$. Also, for $B$ we have,

$$h(\overline{B}) = \int h(\overline{B}(x)) \, dP(x) = \sum_{i=1}^{3} \frac{1}{3} \cdot \int h(B_{\alpha_i}(x)) dP(x) \tag{4}$$

Since our candidate set is a dense expansion of the original 3 masks, each $\alpha$-cut can be uniquely contained in one of the 3 sub-cuts in B. Therefore, for every sub-cut, by the convexity of a characteristic function minus a bunch of sub-characteristic functions,

$$\sum_{i=1}^{3} \frac{1}{3} \cdot \int h(B_{\alpha_i}(x)) dP(x) - \sum_{i=1}^{|A|} p_{\alpha_i} \cdot \int h(A_{\alpha_i}(x)) dP(x) \tag{5}$$

$$= \sum_{i=1}^{|A|} p_{\alpha_i} \cdot \int_{interval(\alpha_i)} [h(B_{\tilde{\alpha}_i}(x)) - h(A_{\alpha_i}(x))] dP(x) \tag{6}$$

$$\geq \sum_{i=1}^{|A|} p_{\alpha_i} \cdot \int_{interval(\alpha_i)} x \, dP(x) > 0 \tag{7}$$

where the first inequality is conducted by Lemma 3.8. $\qquad\square$

## A.2 THEOREM 2

Now we focus on the efficiency of our strategy.

**Definition A.1** ($N_{max}$ and SAM-subregion). *We define the new region $g_{\mathcal{P}}$ added by SAM after applying a prompt $\mathcal{P}$ as the SAM-subregion corresponding to $\mathcal{P}$. We denote the set of all SAM-subregions of an image $x$ as $\mathcal{G}$. For $g \in \mathcal{G}$, let $N_g$ denote the number of other SAM-subregions adjacent to it. Then $N_{max} := \max_{g \in \mathcal{G}} N_g$.*

Therefore, consider the $\mathcal{G}$ as the vertex set and the adjacent relation as the edge set, $\mathcal{G}$ can be regarded as a graph (Bondy and Murty, 2008).

**Lemma A.2** (The Upper Bound of the Minimum Path in a Graph (Bondy and Murty, 2008)). *In a connected graph, if the maximum degree of any vertex is $N_{max}$, and the graph has $|\mathcal{G}|$ vertices, then the length of the minimum path between 2 vertexes $g_i, g_j$ in $\mathcal{G}$ of the graph satisfies the following inequality:*

$$diam(g_i, g_j) \leq \lceil \log_{N_{max}} |\mathcal{G}| \rceil$$

**Assumption A.3** (Relations between $N_{max}$ and $|\mathcal{G}|$). *From experimental results in (Ravi et al., 2025), we assume that the set of SAM-subregions $\mathcal{G}$ is a tree-graph whose root node is in the foreground. Moreover, we assume the depth of every branch in the tree is no more than 3 and $N_{max} \geq 8$. Under this assumption, we conclude that $|\mathcal{G}| \leq (N_{max})^3$. Thus $\log_{N_{max}} |\mathcal{G}| \leq \frac{1}{2} \sqrt[8]{N_{max}}$.*

**Lemma A.4** (Series Expansion Approximation). *For $g_i, g_j$ in a subset of SAM-subregions $\mathcal{G}$, the length of the minimum path between 2 vertexes $\mathrm{diam}(g_i, g_j) \leq \frac{1}{2} \sqrt[8]{N_{max}}$*

*proof of Lemma 1.4.* If there's a single Next Chain Clicker from $g_0$ and clicks on subparts $\mathcal{G}_\rangle$ of $\mathcal{G}$ that contains $g_i$,

$$\mathrm{diam}(g_i, g_0) \leq \log_{N^i_{max}}(|\mathcal{G}_\rangle|)$$

where $N^i_{max}$ is the maximum degree of any vertex on $\mathcal{G}_\rangle$. Similarly, if $g_j$ lies in a single Next Chain Clicker, we have,

$$\mathrm{diam}(g_j, g_0) \leq \log_{N^j_{max}}(|\mathcal{G}_|\|)$$

Then, we have

$$\mathrm{diam}(g_i, g_0) + \mathrm{diam}(g_j, g_0) \leq \log_{N^i_{max}}(|\mathcal{G}_\rangle|) + \log_{N^j_{max}}(|\mathcal{G}_|\|) \tag{8}$$

$$\leq \frac{1}{4} \sqrt[8]{N^i_{max}} + \frac{1}{4} \sqrt[8]{N^j_{max}} \tag{9}$$

$$\leq \frac{1}{2} \sqrt[8]{N_{max}} \tag{10}$$

$\square$

Therefore, we can prove Theorem 2 as follows.

*proof of Theorem 2.* Let $M_0 = \cup\{g_i\}_{i=1}^K$ denote the output mask of the given user's click $(x_0, y_0)$. $\mathcal{G} = \{g_{K+1}, ..., g_N\}$ be the set of erroneous SAM-subregions. Now consider:

(1) If there's a $g_k \in \mathcal{G}$ cannot be searched out by the $\sqrt{N_{max}}$ branches with $\sqrt[4]{N_{max}}$ iterations per branch. Let $\{g_K, g_{k_1}, ...g_{k_t}, g_k\}$ be the shortest connection way to $\mathcal{G}$. Then by Lemma 1.4, $k_t \leq \sqrt[8]{N_{max}}$. Now consider $\mathcal{G}$ as a tree of which the root node is $g_0$. The shortest connection between $\{g_0, g_K, g_{k_1}, ...g_{k_t}, g_k\}$ and other branches is $\leq \frac{1}{2} \sqrt[8]{N_{max}}$. Therefore, we have $\sqrt[8]{N_{max}} > \sqrt[4]{N_{max}}$, which is contradict to the Assumption that $N_{max} > 1$.

(2) Otherwise, the original searching branch can search out the farthest (from the perspective of a connected graph) $g_k$ in $\mathcal{G}$. Then the disambiguous mask $M_0 = \cup\{g_i\}_{i=1}^N$ of the user's intent could be conducted in $\mathcal{G}$. $\square$

## B  PRACTICAL APPLICATIONS

**Why We Regard Reference Instances Easy to Get.** From both the previous work (Liu et al., 2024b; Zhang et al., 2024b; Sun et al., 2024) and the practice, reference instances are acceptable visual prompts. In practice, when dealing with a large number of images to annotate without reference masks, the user can first use the interactive model to get a mask representing the target category.

**Potential Applications.** For large-scale data annotation tasks, SmartSAM significantly reduces both interaction time and operational costs. Additionally, it provides enhanced usability for processing ambiguous images in routine applications.

## C  DATASETS DETAILS

**Statistics of Used Datasets.** As shown in Tab. 11, we report the statistics of the three datasets. *(1) DAVIS:* The DAVIS dataset, used for the interactive segmentation task, consists of 374 keyframes extracted from videos. Following FocSAM (Huang et al., 2024c), all instances in each image are treated as a single instance. Therefore, there are a total of 374 instances. These instances primarily belong to categories such as humans, animals, and vehicles. *(2) PartImageNet:* A total of 2,408 ambiguous images were selected from the test set of PartImageNet. The entire object in each image is treated as the target instance. Since each image in PartImageNet generally contains only one complete instance, there are a total of 2,408 instances. This dataset includes 30 categories in total. *(3) Amb-Occ:* Objects from the 80 base classes in COCO were selected as the target instances. Using

Table 11: The statistics of the three benchmark datasets. *No. Images* denotes the number of images in the dataset. *No. Instances* denotes the number of instances (also the number of masks) in the dataset.

| | DAVIS | PartImageNet | Amb-Occ |
|---|---|---|---|
| No. Images | 374 | 2408 | 2744 |
| No. Instances | 374 | 2408 | 4470 |
| Categories | - † | 30 | 80 |
| Ambiguity | - | $\checkmark$ | $\checkmark$ |

† We follow FocSAM in treating all instances in each image of DAVIS as a single instance, and therefore do not count categories.

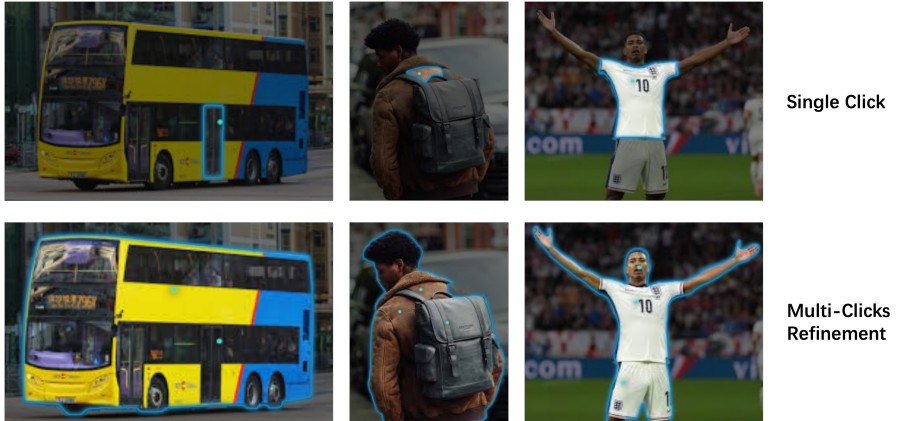

Figure 6: Failure cases of SAM dealing with ambiguous input. Users typically have to add more accurate clicks to get their target masks.

annotations from LVIS and COCO, a dataset comprising 2,744 images and 4,470 instances was constructed. For more details, see the next subsection.

**Why and How We Construct Amb-Occ.** We construct Amb-Occ based on the observation that larger instances are often composed of sub-instances (*i.e.,* a man with a backpack). As shown in Fig. 6 (examples generated using the web demo of SAM), refining the segmentation of whole instances typically requires more iterations of human clicks. To construct the dataset, we selected the LVIS (Gupta et al., 2019) and COCO (Lin et al., 2014) datasets, as they share the same image assets but LVIS provides more detailed annotations. Specifically, we chose the categories in COCO as the target instances and applied a coarse filtering process to identify images where the segmentation mask of one instance is fully contained within a larger mask. Subsequently, we manually refined the dataset, resulting in 2,744 images and 4,470 instance pairs (where one mask contains another). Examples of these are shown in Fig. 7.

# D MORE DETAILS ON THE WORKING MECHANISM

## D.1 DETAILS IN SMARTSAM

As shown in Fig. 8, the Priori Initial Sampler generates a number of additional clicks (9 in our settings). Each click generates 3 masks using the multi-mask mode of SAM, resulting in a set of initial masks (in our settings, $3 \times (9 + 1) = 30$ masks). Subsequently, the Next Chain Clickers iteratively refine the initial masks by adding clicks (3 iterations in our settings). Finally, the agent computes the cosine similarities between the feature of the reference instance and the candidate masks. The top-1 candidate with the highest similarity is selected as the output.

## D.2 A MORE COMPREHENSIVE INVESTIGATION ON OUR BASIS OBSERVATION

**Why We Use Simulator Points.** In interactive segmentation, model evaluation is not performed using actual human clicks. Following (Liu et al., 2023; Huang et al., 2023; 2024c), most interactive segmentation methods employ a click simulator to mimic human clicking behaviors during evaluation.

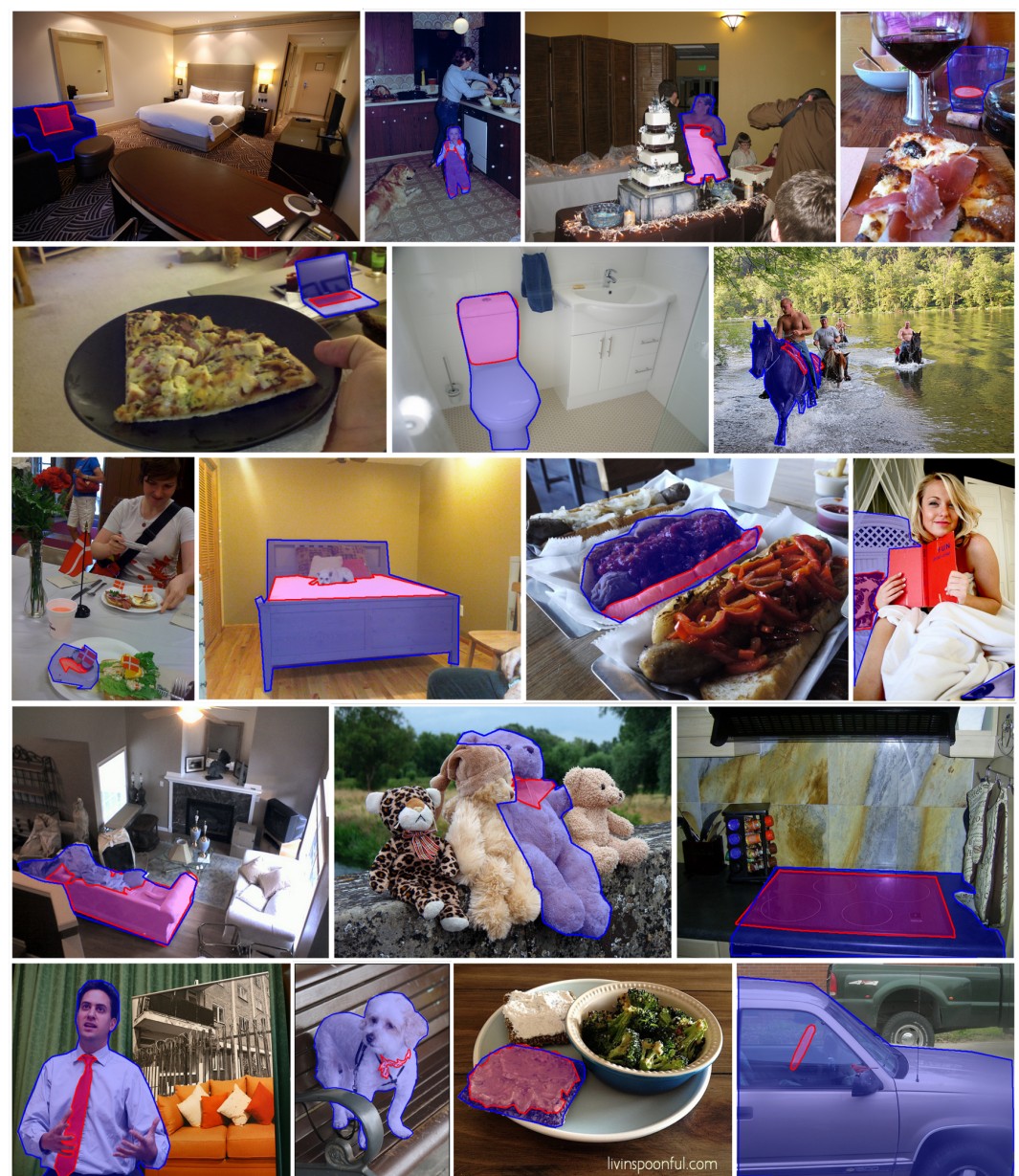

Figure 7: We select from LVIS and COCO datasets to build our Amb-Occ Dataset.

Specifically, this simulation strategy selects a "center" point from erroneous areas, located at a certain distance from the boundary, to simulate human refinement actions. This center point represents the statistical average of human click locations.

**The Distribution of the Logits of Simulated Oracle Points.** We report the logit values of the ambiguous parts and the entire distribution. As shown in Fig. 9, the left panel presents the box plot of the logits for the ambiguous parts, while the right panel shows the entire distribution. We observe a difference between the distributions: the logit values of the ambiguous areas are closer to the segmentation threshold, indicating that these regions correspond to the uncertain areas discussed in the main paper.

### D.3 A THOROUGH ANALYSIS ON PRIORI INITIAL SAMPLER

**Why the Agent Should Add Initial Prompts.** As shown in Fig. 11, when processing ambiguous

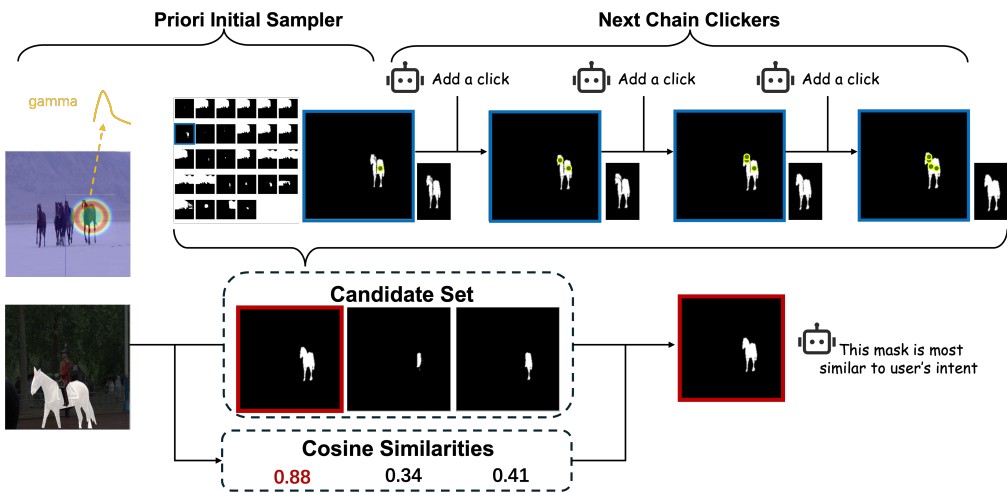

Figure 8: The details of mask generation. Generally, the candidate set is generated through a "search-and-filter" workflow. First, the Priori Initial Sampler samples several masks, and the Next Chain Clickers iteratively refine these masks. Subsequently, SmartSAM computes the cosine similarities between the reference feature and the candidate features, outputting the mask with the highest similarity.

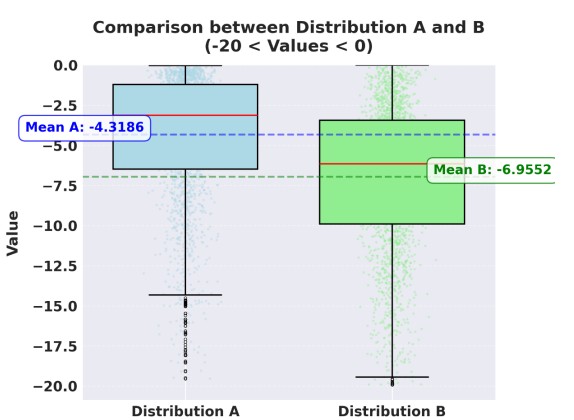

Figure 9: We plot the distributions of the logits for the ambiguous parts (left) and the whole areas (right). The visualized box plots indicate significant differences between the two distributions, with the ambiguous parts exhibiting higher uncertainty.

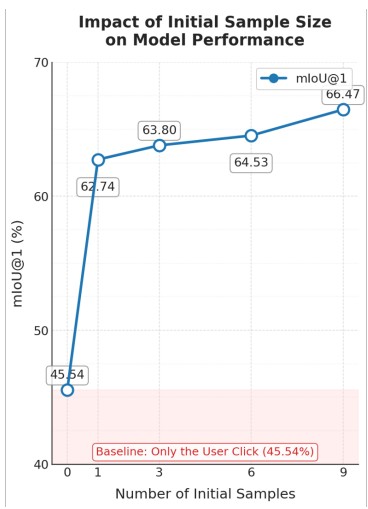

Figure 10: Compared with the baseline SAM, the number of initial prompts exhibits diminishing returns. An excessive number provides limited improvement to performance.

images, a single input may fail to avoid generating unwanted masks. However, by properly sampling additional initial prompts, this issue can be resolved.

**Why Choose Gamma.** SmartSAM using a Gamma distribution performs better than using a normal distribution (see Tab. 12). As shown in Fig. 12, an intuitive explanation for this is that the normal distribution lacks "skewness," which results in oversampling points that are too close to the user click.

**Why the Number of Initial Prompts is Controlled.** Intuitively, increasing the number of initial prompts improves performance. However, as shown in Fig. 10 (with the number of Next Chain Clickers' iterations controlled at 3 except for the user's click), this benefit does not consistently lead to improved performance. Furthermore, an excessive number of branches negatively impacts inference time.

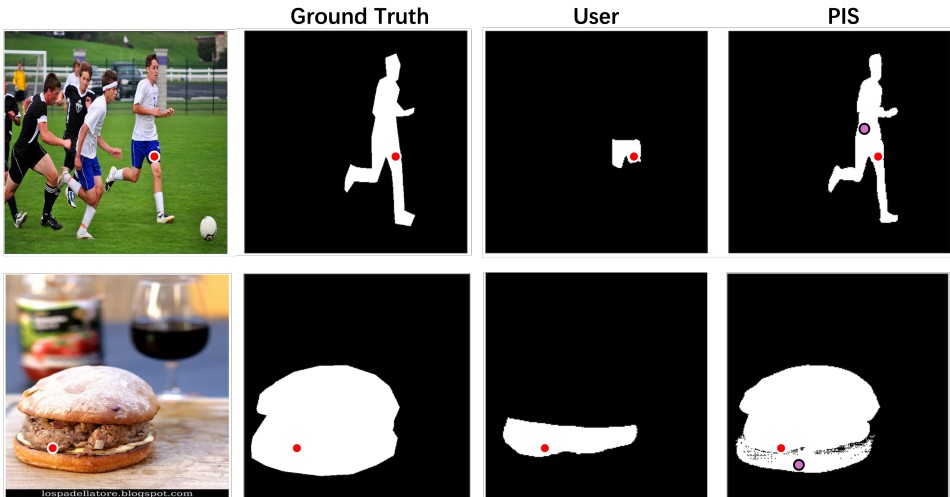

Figure 11: How the Priori Initial Sampler (PIS) works: When users click ● ambiguously, PIS searches for more appropriate masks by clicking ● around users' click.

Table 12: By controlling the peaks to 100 and maintaining the same variations, the results show that when the peaks of the PDFs are controlled identically, the Gamma Distribution is a better choice for SmartSAM.

|  | Ori SAM | Gamma | Norm |
|---|---|---|---|
| PartImageNet | 45.53 | **64.53** | 62.42 |
| Amb-Occ | 37.81 | **47.19** | 44.21 |

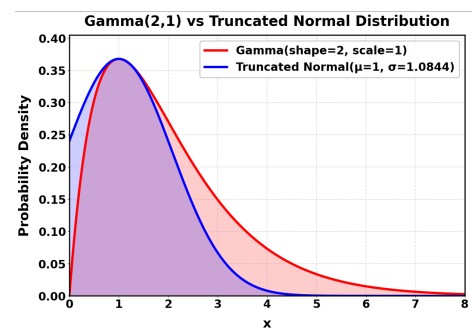

Figure 12: A schematic diagram of a gamma distribution and a normal distribution with the same maximum PDF position.

## D.4 A THOROUGH ANALYSIS ON OUR NEXT CHAIN CLICKERS

**Why Dynamically Change the Range of Uncertain Areas.** From the perspective of definition, "uncertainty" is a concept opposite to "certainty." Thus, the former should be defined in relation to the latter. A certain area in SAM predictions corresponds to the top high (or top low) scores in the logits, meaning that uncertain areas are relatively lower (or higher) than the certain ones. As shown in Fig. 13, the logit distributions of these images span a wide range. Therefore, using an absolute threshold to define "uncertain" is not appropriate.

## D.5 WHY THE LENGTH OF THE CHAIN SHOULD BE CONTROLLED?

As mentioned in Section IV, the quality (measured by *Best IoU* and *mIoU@1*) does not improve as the number of NCC iterations increases. To address this, we present evidence in Fig. 15, which shows that when the number of interactions exceeds 3 clicks, the *mIoU@3* reaches nearly $90\%$ on the PartImageNet and DAVIS datasets, indicating the resolution of ambiguity. One may question whether agent-generated clicks perform as well as oracle human interactions and whether the chain length should be extended. We address this concern in Fig. 14. As shown in the figure, after 3 agent-generated clicks, the ambiguous parts become difficult to distinguish from the background.

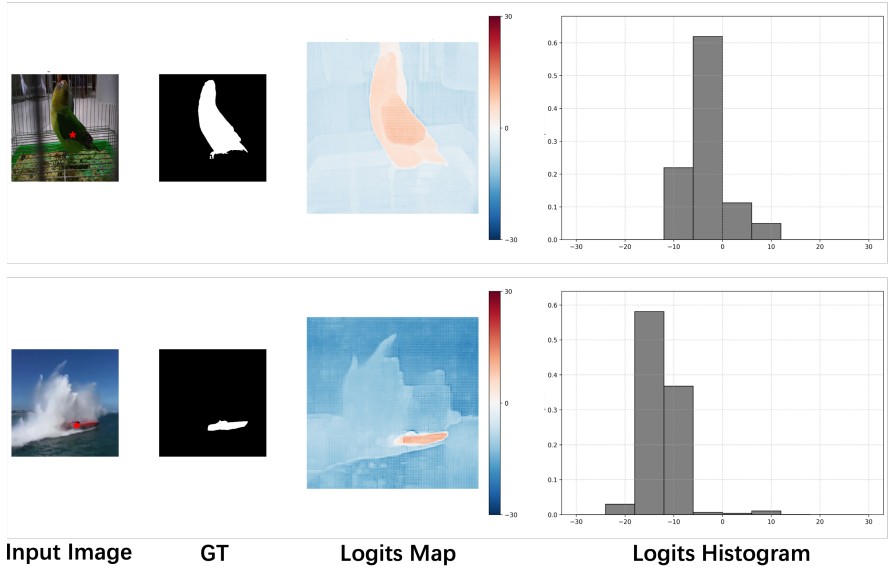

Input Image      GT      Logits Map      Logits Histogram

Figure 13: We plot (from left to right) the input image, ground truth, logits, and their distributions in a histogram. The values of uncertain areas vary at the image level. For the upper image of the parrot, the absolute values of the background are lower than those of the boat image.

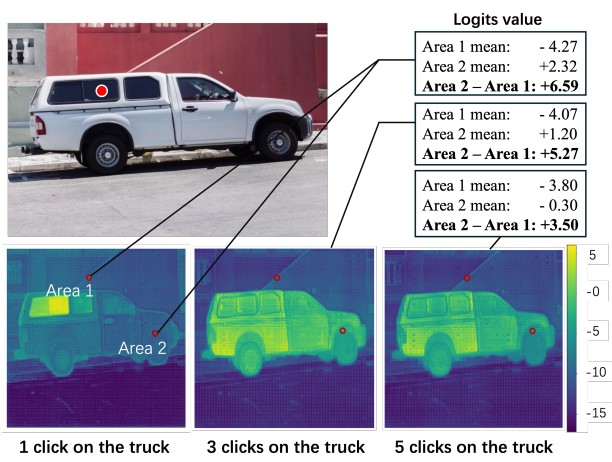

Figure 14: The uncertainty of the ambiguous area does not remain separable as the number of clicks increases. In this figure, when the user adds too many clicks on the windows, the logit scores of the ambiguous area (specifically, other parts of the truck, including Area 2) become indistinguishable from the unrelated background (the red wall, Area 1).

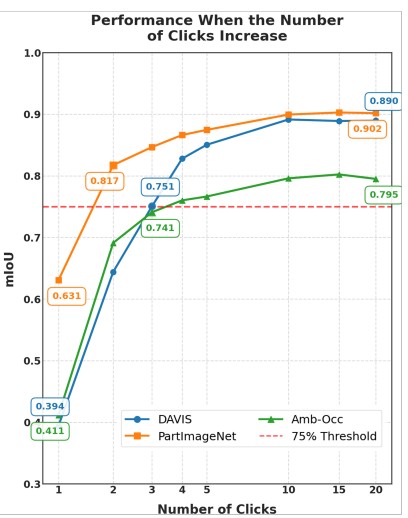

Figure 15: The mIoUs on different datasets as the number of clicks increases are shown. We found that approximately 3 clicks are sufficient to achieve a high level of performance, indicating the resolution of ambiguity.

# E  MORE DETAILS FOR EXPERIMENTS

## E.1  WHY FAILED ON FOCSAM AT RATIO@85

**General Analysis.** One may doubt the main results in Tab. 1 of the main text, where the *Ratio@85* is even worse than the baseline FocSAM. This counterintuitive issue happens because of the candidate selector and the poor separation of FocSAM's logits between the background and the ambiguous region. As shown in Fig. 16, though the original masks generated by the user's click are in the candidate set, the semantic image encoder of the candidate selector mistakes the low-quality masks as the best masks. This is mainly because FocSAM ignored the IoU-head of the original SAM, which

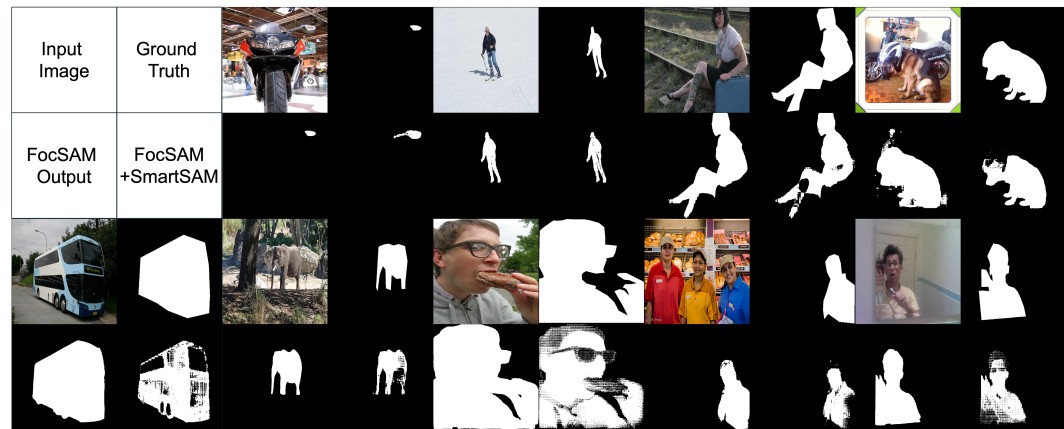

Figure 16: We plot the images where FocSAM outperforms SmartSAM. There are two main cases. First, SmartSAM outputs more reasonable masks (*i.e.*, a complete rearview mirror, a skating man without ski poles, or a dog without a helmet). Second, since FocSAM disables the IoU-Head, it is unable to merge the candidate masks. As a result, for images where FocSAM already performs well, our method occasionally produces low-quality masks.

will be discussed later. Therefore, our strategy failed to increase the performance of the FocSAM on *Ratio@85*.

**Differences Between FocSAM and Other SAMs.** The main difference is that FocSAM disables the

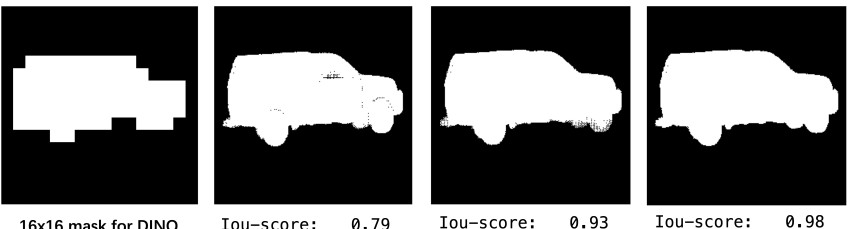

Figure 17: The predicted IoU score can represent the "quality" of the mask. The first example is the interpolated $16 \times 16$ mask obtained through mask-average-pooling. The following three examples share the same low-resolution masks but differ significantly in quality (the second and third contain substantial noise, while the last one is much clearer). Due to the absence of the IoU-Head, FocSAM tends to produce low-quality masks.

IoU-Head module, whereas typical baseline SAMs retain this component. This structure is utilized by SmartSAM to evaluate the quality of the candidate set (see Fig. 17). As a result, the absence of the IoU-Head indeed prevents SmartSAM from outputting high-quality masks.

## F    WHERE WE PLACES THE 1ST CLICKS FOR EXPERIMENTS IN THE REBUTTAL PHASE.

Here is a visulized example (see Fig. **??**) of how the first clicks are placed for experiments in the rebuttal phase.

## G    HOW FOCSAM PREPROCESSES THE DAVIS DATASET.

We follow the preprocess procedure of FocSAM to turn multi-objects DAVIS into single objects (see Fig. 20).

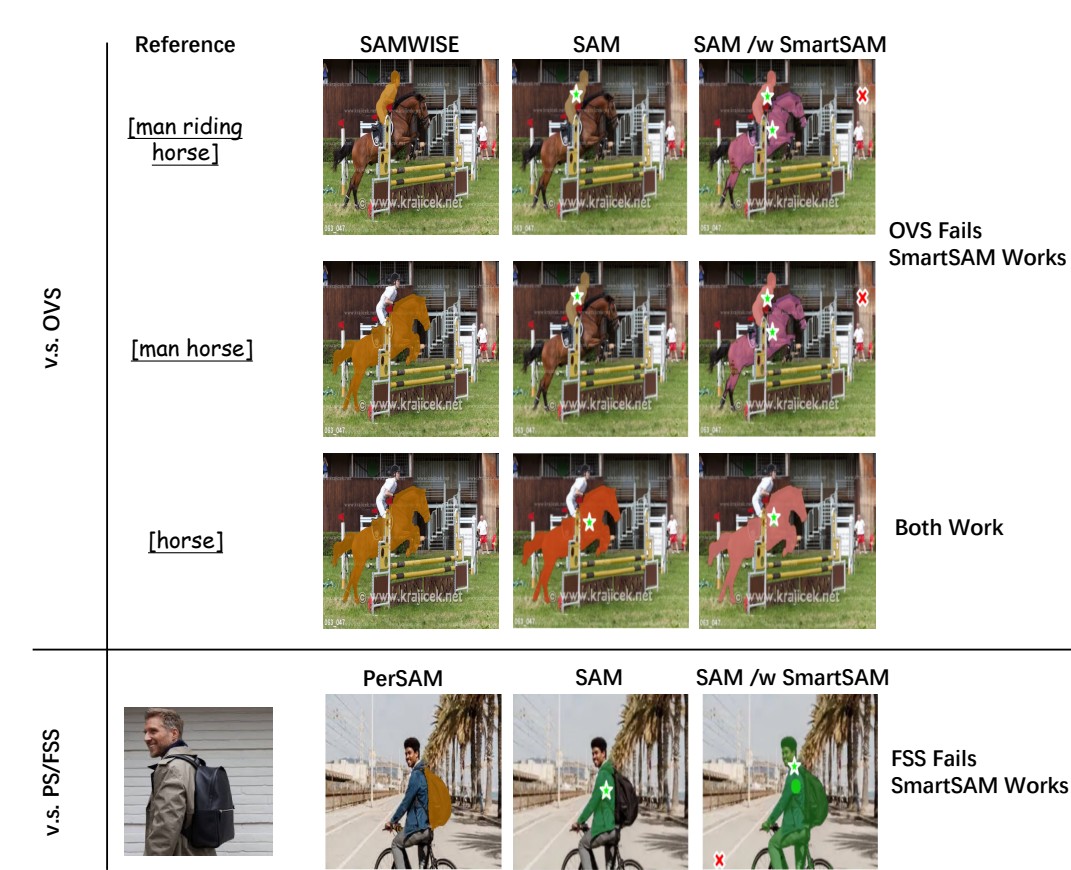

Figure 18: More Visulized Results Compared with FSS/OVS methods. Qualitative comparison between SmartSAM and representative methods across three task families: Interactive Segmentation (IS), Point/Prompt-based Segmentation (PS/FSS), and Open-Vocabulary Segmentation (OVS). In the IS scenario, SAM-based baselines tend to over-focus on partial regions of the object, whereas SmartSAM accurately segments the entire object. In the PS/FSS case (e.g., PerSAM), SmartSAM correctly segments the person riding the bike, while SAMWISE produces an incorrect mask on the black bag. In the OVS case (e.g., SAMWISE), SmartSAM successfully identifies the person riding the horse, but SAMWISE fails to locate the correct target due to its lack of support for human-click prompts.

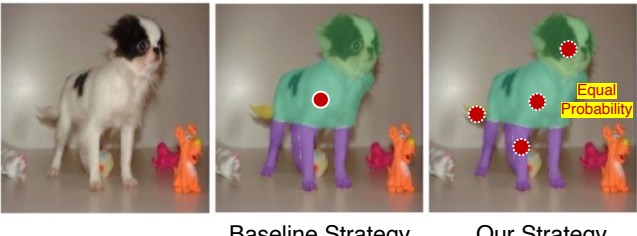

Figure 19: How we simulate real-world human 1st clicks.

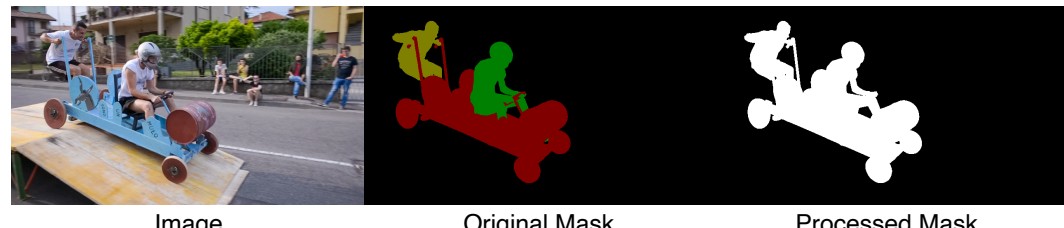

|     |     |     |
| --- | --- | --- |
| Image | Original Mask | Processed Mask |

Figure 20: We merge multi-objects DAVIS into single objects.

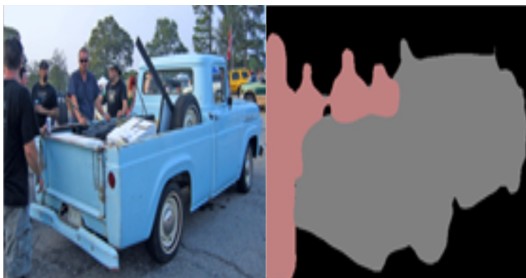

**GT of FSS benchmarks**

Figure 21: FSS benchmarks are not suitable for evaluating IS methods. IS focus on segmenting a single object, while in FSS benchmarks there's always multi-objects of a single category in the image.

