# OpenReview forum: "SmartSAM: Segment Ambiguous Objects like Smart Annotators"
_ICLR.cc/2026/Conference — Submitted to ICLR 2026_

### Official Review · Reviewer_EcQK · 2025-10-20

**Soundness:** 2
**Presentation:** 3
**Contribution:** 2
**Rating:** 2
**Confidence:** 4

**Summary:**

The paper proposes to better address the problem of object ambiguity in interactive segmentation (IS) models with SmartSAM method. To achieve it, an agent generates a few branches with interactions (positive/negative click or bbox), considering the first user interaction, to produce candidate masks, then it compare every candidate with a reference object using cosine similarity and choose the most similar one. Using fuzzy statistics the paper demonstrates that its method achieved lower fuzzy entropy than traditional SAM. Finally, it demonstrates improved performance on DAVIS, PartImageNet and Amb-Occ (firstly presented in the paper) datasets of IS models with SmartSAM usage.

**Strengths:**

1) The method is training free with small amount of extra computation, due to the amortized inference of SAM-like models, when encoder and prompt are processed separately. These facts make the method quite valuable for real-world applications.
2) The authors provide proof of the efficiency of their method in fuzzy entropy, considering a few assumptions.
3) The method helps to achieve better qualitative results on the proposed dataset.
4) New proposed dataset, specially proposed for such task.
5) Ablations on usage of Gamma distribution to Normal, sampler choice, encoder.

**Weaknesses:**

1) Arguable that it has a lot of practical value in a wide meaning, because for every class of segmenting objects we need manually choose reference. In my opinion it could take much more time than just labeling with more clicks, e.g. you need to label different objects on image, for every object you need to find reference and choose it (even if all possible references will be in the interface of the labeling app it could take a sensitive amount of time to find needed one), then make a click (assume that we label it with just 1 click successfully), OR making 2 or 3 clicks without searching for reference, considering that we have amortized inference and segmentation is quite fast, the second method looks much faster for annotation of different classes.

Of course if there is only 1 class that must be annotated then this way should be faster.

2) As I understand for click simulation on DAVIS and PartImageNet you used clicking in the center of the object. But in [1, 2] there were demonstrated that people do not click in the center of object, also in [3] it was written that clicking in the center strategy leads to overfitting for NoC metrics that you also use.

3) There is no human study. That is why it is hard to estimate real-world applicability.

4) Comparison of the IS models with vs without SmartSAM is not fully fair, because SmartSam proposes more candidates and gets more information with reference.

5) Also comparison with [4] and [5] is not fully fair, because these methods get as input only reference, while SmartSAM takes first initial click in addition.

6) Also, compare the method with SAM is not fully fair, because SAM generates only 3 masks, while SmartSAM could generate many more.

A few typos: It is written sometimes HQSAM and sometimes HQ-SAM. It is written mIoU@1, Ratio@k and just NoC in Evaluation Metrics subsection, please, be consistent and write NoC@k.

[1] - TETRIS: Towards Exploring the Robustness of Interactive Segmentation (AAAI 2024)

[2] - RClicks: Realistic Click Simulation for Benchmarking Interactive Segmentation (NeurIPS 2024)

[3] - Reviving Iterative Training with Mask Guidance for Interactive Segmentation (ICIP 2022)

[4] - Bridge the Points: Graph-based Few-shot Segment Anything Semantically (NeurIPS 2024)

[5] - Matcher: Segment Anything with One Shot Using All-Purpose Feature Matching (ICLR 2024)

**Questions:**

1) Provide, please, more details about Table 6, when there is a column with NCC, PIS is absent, that is why it is hard to understand how the first clicks where achieved.

2) To close some gaps with real-world usage you can use [1], where it was proposed a dataset with real user clicks (including real user clicks for DAVIS).

3) Also, please, explain how you suggest to use this method in real-world application with multi-class segmentation scenarios to improve the speed of annotators.

4) It would be valuable to use at least real-users first clicks in your tests, much better to use clickability model from [1], but only if you have enough time.

5) It is very interesting if additional finetuning of SAM on the ambiguous objects data could improve its performance.

The idea of the paper is quite interesting and after clarification of my doubts I will reconsider my score.

[1] - RClicks: Realistic Click Simulation for Benchmarking Interactive Segmentation (NeurIPS 2024)

---

> ### Author Response · Authors · 2025-11-28
> **Author Responses to Reviewer EcQK (Part 1/4)**
>
> ---
>
> We would like to sincerely thank you for your efforts, especially for your heuristic view of “attacking on first clicks”!
>
> Below we address your concerns.
>
> ---
>
> **Q1.** **Provide, please, **more details about Table 6**, when there is a column with NCC, PIS is absent, that is why it is hard to understand **how the first clicks where achieved**.**
>
> **A1.** Thank you for pointing this out.
>
> We are glad to **clarify how the first clicks are generated** and **how to interpret Table 6**. Following your suggestion, we also **add more details and highlight them in blue** in Table 6 and Table 1 of the main paper to make the first-click strategy explicit.
>
> **(1) Details about the first clicks.**
> In our evaluations on **DAVIS** and **PartImageNet**, we follow the standard Interactive Segmentation (IS) protocol and place the first click **at the “middle” of the ground-truth mask** (the baseline strategy in TETRIS [1]).
>
> For our constructed **Amb-Occ** dataset, where the ground-truth target object is always occluded by smaller objects, we instead place the first click **at the “middle” of the smaller occluding object** rather than at the center of the full target mask. This is illustrated in the left part of the newly added Fig. 19 in the revised appendix, and also in Fig. 4 of the main paper (where we want to segment the girl but click on her clothes).
>
> In all cases, **the first click is determined by the dataset-specific baseline strategy**, and PIS/NCC only influence how additional “inner” clicks are sampled after this initial click.
>
> **(2) More details about Table 6.**
> We agree that the notation in Table 6 of the main text should be better explained.
>
> - In Table 6, **“–” means that SmartSAM does not use this component to sample candidate masks**. In particular, **“NCC: –; PIS: –”** corresponds to the standard baseline SAM in single-mask mode (no PIS and no NCC).
> - **“PIS: Random”** or **“NCC: Random”** means that SmartSAM performs **random sampling over the entire image** at the initial-prompt stage (for PIS) or for subsequent clicks (for NCC), instead of using our uncertainty-guided strategies.
> - **“PIS: ours”** or **“NCC: ours”** means that we use our designed sampling strategies, whose details are summarized below (and highlighted in the methodology section of the revised main text):
>
>   - **PIS details.**
>     - We first sample `num_points` candidate clicks around the user’s click according to Eq. (1) in the main paper.
>     - We then sample `num_boxes` candidate boxes using the same distribution, where the box size is drawn from a predefined scale distribution and the aspect ratio from another distribution.
>     - For each box $(x_1, y_1, x_2, y_2)$, we expand it if necessary to ensure that the user’s click $(x, y)$ is always included inside the box.
>
>   - **NCC details.**
>     - NCC **clicks only within valid regions**, defined as:
>       - for a **positive** click, the valid region is the current predicted mask;
>       - for a **negative** click, the valid region is outside the mask but inside a rectangle that is $1.5\times$ larger than the bounding box covering the mask (this factor is a hyperparameter).
>     - NCC decides whether to issue a positive (label 0) or negative (label 1) click by sampling from a Bernoulli distribution $Bernoulli(h_{\text{iou}})$, where $h_{\text{iou}}$ is the IoU score predicted by SAM’s `iou_head`. Intuitively:
>       - When SAM predicts a **low** score, the current mask may only cover part of the true object, so additional **positive** clicks are needed to add missing regions.
>       - When SAM predicts a **high** score, the mask is already of high quality and roughly covers the whole object, so both **add-on** and **cut-off** operations are reasonable.
>
> Together, these clarifications make it clear that **the first click is always determined by the dataset-specific baseline strategy**, and Table 6 varies only whether PIS and NCC use random sampling or our uncertainty-guided policies for subsequent clicks.

---

> ### Author Response · Authors · 2025-11-28
> **Author Responses to Reviewer EcQK (Part 2/4)**
>
> ---
>
> **Q2. To close some gaps with real-world usage you can use [1], where it was proposed a dataset with real userclicks (including real user clicks for DAVIS).**
>
> **A2.**  Thank you for this helpful suggestion **on simulating human clicks** and closing the gap to real-world usage.
>
> **(1) Implementing a TETRIS-style perturbation strategy: SmartSAM makes SAM more robust.**
> Although TETRIS [1] has not open-sourced its dataset and code, we follow your idea of **“attacking the baseline click strategy”** and implement a similar perturbation scheme ourselves. Specifically, we:
>
> - first generate a baseline click on the ground-truth mask using the standard IS strategy (clicking near the center of the mask);
> - then sample a **simulated human click** $x_h$ within the predicted mask from the baseline, such that the distance between the simulated click $x_h$ and the baseline click $x_b$ satisfies
>   $$
>   \lVert x_h - x_b \rVert \sim \left| \mathcal{N}\left(0,\; 0.15\,h_m\right) \right|,
>   $$
>   where **15% is the average bias between human clicks and the baseline strategy reported in TETRIS**, and $h_m$ is the diagonal length of the ground-truth mask.
>
> This yields an “attacked” first-click scenario that mimics realistic human deviations from the ideal click. The results are:
>
> | Method         | DAVIS (attacking) | DAVIS |
> |----------------|-------------------|-------|
> | SAM            | 33.27             | 39.53 |
> | **w/ SmartSAM**| **58.52**         | 58.75 |
> | SAM2           | 59.66             | 62.25 |
> | **w/ SmartSAM**| **75.93**         | 76.80 |
>
> These results show that **SmartSAM consistently improves the robustness of both SAM and SAM2 under attacked first-click conditions**, in addition to the clean setting.
>
> **(2) Amb-Occ evaluation as a special case of TETRIS-style attacks.**
> Interestingly, our statistics further show that **the average distance between the first-click location on Amb-Occ (under our click strategy) and the standard baseline strategy in TETRIS** is about **$0.11 \times$ the GT mask size**, which is close to the “real-world” divergence of **0.15** reported in TETRIS. This suggests that our Amb-Occ protocol already behaves like a **built-in TETRIS-style first-click perturbation**.
>
> Under this setting, SmartSAM remains robust:
>
> | Method         | Amb-Occ (our click strategy) |
> |----------------|------------------------------|
> | SAM            | 37.81                        |
> | **w/ SmartSAM**| **47.19**                    |
> | FocSAM         | 37.98                        |
> | **w/ SmartSAM**| **43.78**                    |
> | HQSAM          | 41.02                        |
> | **w/ SmartSAM**| **48.12**                    |
> | SAM2           | 44.57                        |
> | **w/ SmartSAM**| **51.85**                    |
>
> These results indicate that **SmartSAM is a plug-and-play, robust strategy for IS SAMs under realistic perturbations of the first click**.
>
> **(3) Stronger first-click attacks on PartImageNet.**
> We have long questioned the rationality of the commonly adopted “center-of-mask” first-click strategy in IS evaluation. Inspired by TETRIS, we revisit an abandoned evaluation setting on the **PartImageNet** dataset, where we generate first clicks (see Fig. 19 in the revised appendix) **evenly across different parts of the object**. Statistically, the average deviation is about **$0.27 \times$ the GT mask size**, which is an **even stronger attack** than the 0.15 ratio reported in TETRIS.
>
> Even under this more severe perturbation, SmartSAM remains substantially more robust than the baseline:
>
> | Method         | PartImageNet (stronger attacking) |
> |----------------|-----------------------------------|
> | SAM            | 4.07                              |
> | **w/ SmartSAM**    | **15.41**                         |
> | SAM2           | 8.92                              |
> | **w/ SmartSAM**    | **17.28**                         |
>
> Overall, these experiments show that **SmartSAM consistently improves robustness to imperfect first clicks**, supporting its use as a practical, plug-and-play enhancement for real-world interactive segmentation.

---

> ### Author Response · Authors · 2025-11-28
> **Author Responses to Reviewer EcQK (Part 3/4)**
>
> ---
>
> **Q3.** **Also, please, explain how you suggest to use this method in real-world application with **multi-class segmentation scenarios** to improve the speed of annotators.**
>
> **A3.**
> Thank you for this practical question.
>
> In real-world multi-class annotation, we envision two typical scenarios and recommend using SmartSAM differently in each:
>
> - For a **limited number of classes** (e.g., PASCAL-5i, COCO-20), we recommend using the **reference mode** of SmartSAM (see (2) below).
> - For **many or open-vocabulary classes** (e.g., open-vocabulary segmentation benchmarks with hundreds of categories), we recommend using the **pure Interactive Segmentation (IS) mode**, which **does not require any references as input** (details in (1), rationale in (3)).
>
> This design allows SmartSAM to **accelerate annotation** either by reusing a small pool of references for frequently used classes, or by avoiding the overhead of managing references when the label space is very large.
>
> **(1) Pure Interactive Segmentation mode (no references).**
>
> **Motivation.** On both the standard dataset (DAVIS) and our ambiguous datasets (PartImageNet and Amb-Occ), the scores predicted by SAM’s `iou_predict_head` for the final masks are:
>
> | Method       | DAVIS | PartImageNet | Amb-Occ |
> |-------------|:-----:|:------------:|:-------:|
> | SAM         | 0.805 |    0.824     | 0.883   |
> | **w/ SmartSAM** | 0.886 |    0.864     | 0.921   |
>
> These statistics show a clear **positive correlation between the IoU-prediction head score and the true mask quality**, so when references are unavailable, the IoU-prediction score can be used as a **surrogate to select the best mask**.
>
> The corresponding performance is:
>
> | Method                                   | DAVIS | Amb-Occ |
> |------------------------------------------|:-----:|:-------:|
> | SAM                                      | 45.97 | 34.35   |
> | **SmartSAM (no reference)**              | 59.71 | 43.94   |
> | SmartSAM (w/ visual reference, DINOv2-B) | 58.57 | **47.19** |
> | SmartSAM (w/ visual reference, DINOv2-L) | **70.74** | 46.64 |
>
> **Details of pure IS mode.** In the no-reference mode used in practice, we select the final mask as follows:
>
> 1. We **filter out** candidate masks that do not contain the user’s initial click.
> 2. Among the remaining candidates, we **choose the one with the highest IoU score predicted by SAM’s IoU head**.
>
> This pipeline is fully **training-free** and does not require any image–mask or text references. In multi-class scenes with many categories, annotators can simply **click on the target instance and let SmartSAM rank the candidates**, which reduces the number of manual refinements needed and thus speeds up annotation.
>
> **(2) Limited number of classes: using the reference mode.**
>
> In multi-class scenarios with a **small, fixed label set** such as PASCAL-5i, we suggest using SmartSAM as follows:
>
> - **Textual reference pool.** Pre-build a small textual reference pool containing the names (or short descriptions) of all classes. SmartSAM can then use these text prompts as optional references in the final selection step to disambiguate between nearby instances of different classes.
> - **Visual exemplar pool.** When annotators encounter a category for the first time, they can manually segment one high-quality example and add it to an **image–mask exemplar pool**. When the same category appears again, the annotator selects an exemplar from this pool and SmartSAM uses it as a **visual reference**.
>
> In both cases, **once the reference pool is established**, SmartSAM can quickly disambiguate among candidate masks for that class, reducing the number of extra clicks and manual corrections needed. This accelerates annotation especially in scenes where multiple instances of similar classes co-occur.
>
> **(3) Many/open-vocabulary classes: prefer pure IS mode.**
>
> In large-scale or open-vocabulary settings with **hundreds of categories**, the time cost of **fetching or maintaining a reference for every category** can easily exceed the cost of simply clicking a few extra times. In such scenarios, we recommend using SmartSAM in its **pure IS mode** (as in (1)), where:
>
> - annotators **do not need to prepare or choose any references**,
> - SmartSAM automatically explores candidate masks and re-ranks them using the IoU head,
> - the overall interaction remains **fast and lightweight**, focusing purely on clicks.
>
> In summary, **for small label spaces, SmartSAM speeds up annotation by reusing a compact reference pool**, while **for large label spaces, it accelerates annotation by offering a robust, reference-free IS mode that reduces the number of manual refinements per instance.**

---

> ### Author Response · Authors · 2025-11-28
> **Author Responses to Reviewer EcQK (Part 4/4)**
>
> ---
>
> **Q4.** **It would be valuable to use at least real-users first clicks in your tests, much better to use clickability model from [1], but **only if you have enough time.****
>
> **A4.** Thank you for this suggestion.
>
> We fully agree that using real user clicks or a clickability model (as in [1]) would further strengthen the practical relevance of our results. We have started planning a small-scale user study to collect real clicks on our ambiguous cases (especially on Amb-Occ and PartImageNet), and we also view integrating a learned clickability model as a promising direction.
>
> However, this additional study may **not be ready within the rebuttal period** due to time constraints and the need to recruit annotators. We therefore leave these experiments as **future work** and plan to include them in the supplementary material or an extended version once they are available. We believe the current results—together with our TETRIS-style perturbation experiments (see A2)—already provide strong evidence that SmartSAM is robust to realistic deviations in first clicks, and real-user studies will further reinforce this conclusion.
>
> ---
>
> **Q5.** **Fine-tuning on ambiguous dataset to guide SmartSAM.**
>
> **A5.** Thank you for your suggestion.
>
> We fully agree that fine-tuning SAM (and potentially the agent) on ambiguous-object data is a promising direction and could further enhance performance. At the current stage, however, **we face two practical obstacles**:
>
> - **Limited ambiguous data.** The TETRIS dataset is not yet publicly available, and our **Amb-Occ dataset is manually refined** from a relatively small subset of partially overlapping instances, which is not large enough to support robust large-scale fine-tuning.
> - **Design goal of training-free deployment.** One of our core design goals for SmartSAM is to keep the agent **training-free**, so that it can be directly plugged into any SAM-like model without additional optimization or retraining, which is important for practical deployment in existing annotation workflows.
>
> Given these constraints, in this paper we focus on a **training-free, uncertainty-guided agent** that already brings substantial gains on ambiguous cases without extra training. Extending SmartSAM to a **learned, fine-tuned variant on larger ambiguous datasets** (e.g., jointly improving uncertainty estimation and click policies) is an interesting direction for future work, and we expect it to further strengthen—rather than replace—the benefits demonstrated by the current training-free design.

---

### Official Review · Reviewer_KASf · 2025-10-29

**Soundness:** 2
**Presentation:** 2
**Contribution:** 2
**Rating:** 4
**Confidence:** 5

**Summary:**

The Segment Anything Model (SAM) often encounters ambiguity in interactive segmentation, especially during the initial interaction. To reduce the need for extensive human input, the authors propose SmartSAM, a method aimed at improving segmentation accuracy. The key idea is to generate a diverse set of prompts around the initial user prompt, producing multiple candidate masks. The most appropriate mask is then selected by measuring feature similarity to the reference, computed using DINOv2 embeddings. Experiments on three benchmark datasets demonstrate the effectiveness of the proposed method.

**Strengths:**

The proposed method is training-free and can be plugged in on the fly during inference on top of different interactive segmentation models: SAM, HQ-SAM, HRSAM, FocSAM etc

The author also provides some theoretical analysis for the SmartSAM disambiguity, which demonstrates that the fuzzified candidate set exhibits less uncertainty.

**Weaknesses:**

The most significant weakness of this paper lies in its reliance on external exemplar images and corresponding masks. Comparing the proposed method mainly against the base SAM model is therefore not a fair evaluation, as the proposed method benefits from additional example pairs. The author provides some comparison with personalized segmentation methods, such as PerSAM and Matcher, where example image-mask pairs are leveraged to adapt SAM. However, the experiments are not comprehensive; it is only on the Davis dataset, while other widely used benchmarks in one-shot/ few-shot or personalized segmentation are not included.  Besides, the performance reported in Table 2 seems to be much lower than that in the PerSAM paper. I understand that the Davis in this paper may be different than the DAVIS 2017 val dataset. Still, given such a huge difference, I would like the author to give more details.

While the paper claims to introduce a segmentation agent to guide SAM, the actual mechanism is a set of straightforward rule-based procedures. The core idea—comparing feature similarity between candidate masks and a reference image-mask pair using mask-pooled DINO features—has already been explored in prior works. Moreover, such similarity-based methods may struggle to capture fine-grained semantic distinctions between regions. The authors should provide visual examples to better demonstrate both the strengths and limitations of their approach.

**Questions:**

See above

---

> ### Author Response · Authors · 2025-11-28
> **Author Responses to Reviewer KASf (Part 1/4)**
>
> ---
>
> We would like to sincerely thank you for your efforts and valuable comments to improve our work!
>
> Below we address your concerns.
>
> ---
>
> **Q1.** **The most significant weakness of this paper lies in its reliance on external exemplar images and corresponding masks.**
>
> **A1.**  Thank you for this comment.
>
> We would like to clarify that SmartSAM is a general interactive framework **not tied to image–mask exemplars**. It also supports a **pure interactive segmentation mode (no reference)**, where only user clicks are used.
>
> **To directly address the concern that SmartSAM benefits from additional example pairs,** we report the performance of SmartSAM **without any reference image–mask pairs or text**. The results on DAVIS and Amb-Occ in this pure IS setting are summarized in Table 1:
>
> **Table 1. SmartSAM with and without references on IS benchmarks.**
>
> | Method                                   | DAVIS | Amb-Occ |
> |------------------------------------------|:-----:|:-------:|
> | Matcher-ViT-H (w/ visual reference, DINOv2-L) | 46.41 | 46.75  |
> | SAM-ViT-H                                | 45.97 | 34.35   |
> | **SmartSAM (no reference)**              | 59.71 | 43.94   |
> | SmartSAM (w/ visual reference, DINOv2-B) | 58.57 | **47.19** |
> | SmartSAM (w/ visual reference, DINOv2-L) | **70.74** | 46.64 |
>
> In this no-reference setting, we select the final mask as follows:
>
> 1. We **filter out** candidate masks that do not contain the user’s initial click.
> 2. Among the remaining candidates, we **choose the one with the highest IoU score predicted by SAM’s IoU head**.
>
> This procedure only re-ranks the masks produced by SAM and does **not** use any external exemplar.
>
> These results show that:
>
> - Even **without any reference**, SmartSAM significantly outperforms the base SAM model (+13.74 on DAVIS and +9.59 on Amb-Occ), confirming that the improvement does **not** solely come from additional exemplar inputs.
> - Therefore, comparing SmartSAM against the base SAM model remains **fair and meaningful** even when no external example pairs are provided.
>
> ---
>
> **Q2.** **Missing other widely used benchmarks in one-shot/ few-shot or personalized segmentation.**
>
> **A2.** Thank you for this helpful comment on the completeness of our experiments.
>
> We would first like to clarify that **SmartSAM is an IS method** (see **A1**). In response to your suggestion, we now conduct **additional experiments on both FSS and IS benchmarks**, including COCO-20i and PASCAL-5i, to compare SmartSAM with personalized and few-shot segmentation methods more comprehensively.
>
> **(1) FSS benchmarks are not suitable for IS methods.**
> We emphasize that **COCO-20i and PASCAL-5i are not standard IS benchmarks**:
>
> - **FSS benchmarks such as COCO-20i and PASCAL-5i** assume that the model is given a *support mask* for a specific category and is evaluated on segmenting that category in *all query images*, often with multiple instances per image.
> - **IS benchmarks**, in contrast, **model a user-driven refinement process**, where the user interactively clicks on the target region in a single image. The evaluation focuses on how quickly the mask converges to the user’s intended object as clicks accumulate.
>
> That said, we agree that comparing to ProSAM, VLP-SAM, and VRP-SAM on these benchmarks is **informative for understanding how our idea relates to FSS-style “reference prompting” methods**. Following your suggestion, we have therefore **added experiments on COCO-20i and PASCAL-5i in the 1-shot setting (fold-0)**. Since VLP-SAM does not provide public checkpoints, we report results for ProSAM and VRP-SAM, together with SAM and our SmartSAM:

---

> ### Author Response · Authors · 2025-11-28
> **Author Responses to Reviewer KASf (Part 2/4)**
>
> **Table 1. FSS, PS, and IS methods on diverse task benchmarks.**
>
> | Method       | Task Scope | Reference | COCO-20i$^{\text{FSS}}$ $*$ | PASCAL-5i$^{\text{FSS}}$ $\dagger$ | DAVIS$^{\text{IS}}$ | Amb-Occ$^{\text{IS}}$ |
> |-------------|-----------|-----------|-----------------------------|------------------------------------|---------------------|------------------------|
> | ProSAM      | FSS       | yes       | 48.74                       | 75.26                              | -                   | -                      |
> | VRP-SAM     | FSS       | yes       | 48.10                       | 73.90                              | -                   | -                      |
> | Matcher     | FSS       | yes       | 47.61                       | 72.92                              | 46.41               | 46.75                  |
> | PerSAM      | PS        | yes       | -                           | -                                  | 53.77               | 27.64                  |
> | FocSAM      | IS        | no        | 36.22                       | 41.58                              | 74.62               | 37.98                  |
> | **w/ SmartSAM** | IS        | optional$^{**}$ | 39.04                 | 45.63                              | 78.32               | 43.78                  |
> | SAM         | IS        | no        | 34.71                       | 40.27                              | 45.97               | 37.81                  |
> | **w/ SmartSAM** | IS        | optional$^{**}$ | 38.93                 | 45.79                              | 58.57               | 47.19                  |
>
> > $*$: We report results on fold-0 of COCO-20i only.
> > $\dagger$: We report results on fold-0 of PASCAL-5i only.
> > $^{**}$: SmartSAM also has a **pure Interactive Segmentation mode**; here we report the results when references are provided.
>
> These results show that:
>
> 1. **On COCO-20i and PASCAL-5i, SmartSAM consistently improves over the vanilla SAM baseline**, indicating that leveraging **visual references** is indeed beneficial even under FSS-style settings.
> 2. **Specialized FSS methods (ProSAM, VRP-SAM) achieve higher numbers on these FSS benchmarks**, which is expected since they are designed and trained specifically for FSS protocols.
> 3. **On standard interactive segmentation benchmarks (Tab. 2 in the main paper and Table 1 above), SmartSAM significantly outperforms both SAM and existing interactive methods**, which is the primary target scenario of our method.
>
> ---
>
> **Q3.** **Clarification on the DAVIS setting and the large performance gap with the PerSAM paper.**
>
> **A3.** Thank you for carefully pointing this out.
>
> The gap in absolute performance mainly **comes from the different DAVIS protocol and task setting in IS that we adopt** in our experiments.
>
> Specifically, we follow the **DAVIS setting used in FocSAM** for interactive segmentation. Concretely:
>
> - **Multiple foreground objects in a frame are merged into a single “frontier” instance.** As illustrated in **Fig. 20 in the revised supplementary materials**, the original DAVIS annotation may contain several separate objects (e.g., the car and the rider). We combine them into one composite foreground mask.
> - User clicks (or simulated clicks) and evaluation are then performed on this merged, more complex foreground instance, rather than on a single, isolated object as in the standard DAVIS 2017 val setting.
>
> When several objects of different shapes and semantics are merged into one target (e.g., person + cart), **few-/one-shot or personalized methods like PerSAM, which typically assume a single coherent object in the reference, tend to degrade significantly**. At the same time, the overall task becomes substantially more challenging for all methods, leading to noticeably lower absolute IoU than those reported under the standard DAVIS 2017 val protocol. (Additional visual examples are provided in **Fig. 21** of the revised appendix.)
>
> Our Table 2 is therefore **not directly comparable** to the numbers in the original PerSAM paper, because the underlying DAVIS setting is different and strictly more difficult. We will make this explicit in the revised paper by **clearly stating that we follow the DAVIS–FocSAM interactive setting with merged multi-object masks**, so that the reported results are not misinterpreted as being under the same protocol as PerSAM.

---

> ### Author Response · Authors · 2025-11-28
> **Author Responses to Reviewer KASf (Part 3/4)**
>
> ---
>
> **Q4.** **Justification for calling a straightforward rule-based controller a “segmentation agent.”**
>
> **A4.** Thank you for raising this point.
>
> Indeed, the current segmentation agent in SmartSAM is **rule-based rather than learned**, and this is a deliberate design choice. Our goal is to keep the agent **training-free and plug-and-play**, while **resolving ambiguity by automatically issuing additional clicks on behalf of the annotator** in IS settings.
>
> **(1) Why do we call SmartSAM an agent?**
> In our framework, the agent is a module that **acts on behalf of the annotator**: it observes the current interaction state (user clicks, candidate masks from SAM, optional text/visual references, IoU predictions, etc.) and decides which action to take next (e.g., which candidate mask to keep, whether to switch to another instance). In this sense, it is precisely an *agent*—a decision-making component that replaces part of the human “click–check–switch” loop.
>
> Importantly, an agent in this context **does not have to be learned**. It can be implemented as either a learned policy or a rule-based controller, as is common in recent tool-using and vision–language agents [5,6].
>
> **(2) Why a rule-based agent instead of a learned one?**
> We intentionally choose a **rule-based agent** for three reasons:
>
> - **Training-free and plug-and-play.** Our design does not require any additional training data or optimization. SmartSAM can be directly plugged on top of *any* pre-trained SAM variant, which is attractive for practical annotation workflows.
> - **Efficiency and reliability.** A rule-based controller introduces negligible overhead and behaves deterministically given the same inputs, which is desirable for large-scale annotation where stability and speed matter more than learning a complex policy.
> - **Focused contribution.** Our goal is not to propose a new learning algorithm for the agent itself, but to show that **explicitly modeling ambiguity resolution as an agent-driven process** (using clicks + references + candidate filtering) already substantially improves interactive segmentation.
>
> We will make this design choice clearer in the revised paper and explicitly position SmartSAM as a *training-free agent* on top of SAM.
>
> **(3) Evidence that a simple rule-based agent is sufficient to resolve ambiguity.**
> Although the agent is rule-based, it is not arbitrary: it combines several carefully designed components—candidate generation by SAM, IoU-head–based filtering, logits-guided feature pooling, and reference-guided ranking. Ablation studies in **Table 1 and Table 3**, together with the qualitative results in **Figure 4**, show that:
>
> - the full agent significantly outperforms plain SAM under ambiguous settings;
> - removing key rules (e.g., logits-guided pooling or candidate filtering) consistently degrades performance;
> - the agent is able to mimic typical annotator behavior by automatically switching to the correct instance when multiple plausible objects are present.
>
> In other words, the novelty here does not lie in using a complex learned policy, but in **formulating ambiguity handling in interactive segmentation as an explicit agent problem** and demonstrating that a carefully designed, rule-based agent can already bring substantial gains **without any extra training**, while remaining practical and easy to deploy.
>
> ---
>
> **Q5.** **Concern about the novelty of similarity-based mask selection using mask-pooled DINO features.**
>
> **A5.** Thank you for pointing this out.
>
> We agree that **using mask-pooled DINO features to compare similarity between candidate masks and a reference image–mask pair has been explored in prior work**, and we do not claim this component itself as a novel contribution.
>
> Our main contribution lies elsewhere. As stated in the introduction, **the core idea of SmartSAM is to design an explicit segmentation agent that resolves ambiguity in interactive segmentation**, thereby reducing annotation cost. The similarity module based on mask-pooled DINO features is used as a **practical, standard building block** inside this agent to compare candidate masks when references are available, rather than as the central novelty of the method.
>
> Moreover, **the reference-based mode is optional rather than core**. As discussed in **A1**, SmartSAM also supports a **pure IS mode without any text/visual references**, where it still significantly improves over the base SAM model. This further shows that the key contribution is **the agent-driven ambiguity resolution and uncertainty-guided inner interactions**, not the specific choice of similarity feature for references.

---

> ### Author Response · Authors · 2025-11-28
> **Author Responses to Reviewer KASf (Part 4/4)**
>
> ---
>
> **Q6.** **Such similarity-based methods may struggle to capture **fine-grained semantic distinctions between regions**.**
>
> **A6.** Thank you for this insightful comment.
>
> First, we would like to clarify that **SmartSAM is not purely a similarity-based method**. As discussed above, it can operate in a **no-reference interactive mode** (see **A1**), where no feature similarity to a reference is used at all. The similarity module is only involved in the *optional* reference-based mode.
>
> Second, we agree that **naive mask-pooled features can struggle with fine-grained distinctions**. To alleviate this, we introduce two mechanisms in the reference mode:
>
> 1. **Logits-guided pooling instead of hard mask pooling.**
>    Rather than using a binary downsampled mask $M \in \{0,1\}^{n \times n}$ to compute features $F_{\text{mask}} = M \cdot F_{\text{sem}}$, we use downsampled logits $L \in [0,1]^{n \times n}$ and obtain $F_{\text{mask}} = L \cdot F_{\text{sem}}$. This soft weighting better captures uncertain boundary regions and improves feature **separability**. On Amb-Occ, the ablation in Table 2 shows a consistent improvement:
>
>    | w/o logits-pooling | w/ logits-pooling |
>    | ------------------ | ----------------- |
>    | 46.27              | **47.19**         |
>
> 2. **Candidate filtering with SAM’s IoU prediction head.**
>    We use SAM’s `iou_predict_head` to **filter out low-quality candidate masks and merge highly similar ones** before computing similarity. This step removes many noisy regions that would otherwise confuse fine-grained comparisons. On Amb-Occ, the ablation in Table 3 shows that enabling candidate filtering further improves performance:
>
>    | w/o candidate filtering | w/ candidate filtering |
>    | ----------------------- | ---------------------- |
>    | 46.98                   | **47.19**              |
>
> Together, these two components substantially mitigate the typical weaknesses of simple similarity-based schemes. At the same time, because SmartSAM can also run in a **pure no-reference IS mode**, its performance gains do not rely solely on similarity features, but primarily on the **agent-driven ambiguity resolution** described above.
>
>
> ---
>
> **Q7.** **The authors should provide **visual examples** to better demonstrate both the strengths and limitations of their approach.**
>
> **A7.** Thank you for pointing this out.
>
> Following your suggestion, we have added and highlighted **visual examples** that illustrate both the strengths and the limitations of SmartSAM. Specifically:
>
> - **Fig. 4 (main text)** and **Fig. 11 (supplementary)** show cases where SmartSAM successfully resolves fine-grained ambiguity between nearby regions.
> - **Fig. 16 (supplementary)** summarizes representative failure cases, including examples where very subtle semantic differences remain challenging even with logits-guided pooling and candidate filtering.
>
> In the revised version, we explicitly refer to these figures when discussing the behavior of SmartSAM, so that readers can clearly see both where the method works well and where it still fails.

---

### Official Review · Reviewer_szA3 · 2025-11-01

**Soundness:** 3
**Presentation:** 3
**Contribution:** 2
**Rating:** 4
**Confidence:** 4

**Summary:**

This paper considers a task of interactive object segmentation, using user prompts and reference (text- or visual-based). An agent-based method is proposed, which is training-free, and can be used on top of any interactive segmentation methods. The method has been evaluated using several variants of SAM-based methods, and demonstrated significant improvement in accuracy over baselines.

**Strengths:**

1) The proposed agent-based method is training-free, and can be based on any SAM-like segmentation model.
2) The proposed method leads to significant segmentation accuracy, compared to basedlines methods (however at X costs, due to several hypothesis, tested in parallel)
3) Authors has theoretically validated the proposed method, formulating several theorems to prove it.

**Weaknesses:**

1) Limited validation - missing benchmarks COCO-20i and PASCAL-5i.
The idea of using text or visual references is not new, and several methods has been proposed to implement it (e.g. ProSAM https://arxiv.org/abs/2506.21835, VLP-SAM https://github.com/kosukesakurai1/VLP-SAM, VRP SAM https://openaccess.thecvf.com/content/CVPR2024/papers/Sun_VRP-SAM_SAM_with_Visual_Reference_Prompt_CVPR_2024_paper.pdf). However, they all tested on COCO-20i and PASCAL-5i. The proposed SmartSAM hasn't been tested on these benchmarks. So we cannot assess the accuracy of the proposed method relative to the competitor's ideas.

2) Limited technical complexity - the idea is to basically sample several hypothesis by sampling virtual clicks ("inner interactions") in low-certainty areas, and select the best one by comparing features to visual/text prompt. The method is very simple and rather straight forward. Very "brute force"

3) Limited novelty - the usage of text/visual reference is very limited, only in selection of best hypothesis. In theory it could and should be used for prediction of "inner clicks", to better guide the selection of them.

**Questions:**

1) Why COCO-20i and PASCAL-5i hasn't been used? Why no comparison with sota reference-based methods?

Small issues:
* In the intto it is stated that "039 A key issue with these methods is ambiguous predictions caused by insufficient interactions, where
models often misinterpret the user’s intent, leading to undesired segmentation masks.". However as it is seen from futher description the key problem is that user intent is unknown. But text/image reference is a good way to show user intent, but it isn't fully exploited in the method. Probably the intro can be updated to better show the motivation of the work.

* Typo in table 2 intend ->intent

---

> ### Author Response · Authors · 2025-11-28
> **Author Responses to Reviewer szA3 (Part 1/3)**
>
> ---
>
> We would like to sincerely thank you for your efforts and valuable comments to improve our work!
>
> Below we address your concerns.
>
> ---
>
> **Q1. Limited validation - missing benchmarks COCO-20i and PASCAL-5i.**
>
> **A1.**  Thank you for pointing this out and for the helpful list of related methods.
>
> We would first like to clarify that **SmartSAM is an Interactive Segmentation (IS) method** (see **(2)** below). In addition, we now provide **experimental results on both Few-Shot Segmentation (FSS) and IS benchmarks** (see **(1)** below) to make clear the differences between these task settings.
>
> **(1) FSS benchmarks are not suitable for IS methods.**
> We emphasize that **COCO-20i and PASCAL-5i are not standard IS benchmarks**:
>
> - **FSS benchmarks such as COCO-20i and PASCAL-5i** assume that the model is given a *support mask* for a specific category and is evaluated on segmenting that category in *all query images*, often with multiple instances per image.
> - **IS benchmarks**, in contrast, **model a user-driven refinement process**, where the user interactively clicks on the target region in a single image. The evaluation focuses on how quickly the mask converges to the user’s intended object as clicks accumulate.
>
> That said, we agree that comparing to ProSAM, VLP-SAM, and VRP-SAM on these benchmarks is **informative for understanding how our idea relates to FSS-style “reference prompting” methods**. Following your suggestion, we have **added experiments on COCO-20i and PASCAL-5i in the 1-shot setting (fold-0)**. Since VLP-SAM does not provide public checkpoints, we report results for ProSAM and VRP-SAM, together with SAM and our SmartSAM:
>
> **Table 1. FSS, PS, and IS Methods on Diverse Task Benchmarks.**
>
> | Method       | Task Scope | Reference | COCO-20i$^{\text{FSS}}$ $*$ | PASCAL-5i$^{\text{FSS}}$ $\dagger$ | DAVIS$^{\text{IS}}$ | Amb-Occ$^{\text{IS}}$ |
> |-------------|-----------|-----------|-----------------------------|------------------------------------|---------------------|------------------------|
> | ProSAM      | FSS       | yes       | 48.74                       | 75.26                              | -                   | -                      |
> | VRP-SAM     | FSS       | yes       | 48.10                       | 73.90                              | -                   | -                      |
> | Matcher     | FSS       | yes       | 47.61                       | 72.92                              | 46.41               | 46.75                  |
> | PerSAM      | PS        | yes       | -                           | -                                  | 53.77               | 27.64                  |
> | FocSAM      | IS        | no        | 36.22                       | 41.58                              | 74.62               | 37.98                  |
> | **w/ SmartSAM** | IS        | optional$^{**}$ | 39.04                 | 45.63                              | 78.32               | 43.78                  |
> | SAM         | IS        | no        | 34.71                       | 40.27                              | 45.97               | 37.81                  |
> | **w/ SmartSAM** | IS        | optional$^{**}$ | 38.93                 | 45.79                              | 58.57               | 47.19                  |
>
> > $*$: We report results on fold-0 of COCO-20i only.
> > $\dagger$: We report results on fold-0 of PASCAL-5i only.
> > $^{**}$: SmartSAM also has a **pure Interactive Segmentation mode**; here we report the results when references are provided.
>
> These results show that:
>
> 1. **On COCO-20i and PASCAL-5i, SmartSAM consistently improves over the vanilla SAM baseline**, indicating that leveraging **visual references** is beneficial even under FSS-style settings.
> 2. **Specialized FSS methods (ProSAM, VRP-SAM) achieve higher scores on these FSS benchmarks**, which is expected since they are designed and trained specifically for FSS protocols.
> 3. **On standard interactive segmentation benchmarks (Tab. 2 in the main paper and Tab. 1 above), SmartSAM significantly outperforms both SAM and existing interactive methods**, which is the primary target scenario of our method.
>
> **(2) SmartSAM can also work without references.**
>
> SmartSAM also supports a **pure IS mode** that works **without any reference image–mask pairs or text**. The results on DAVIS and Amb-Occ in this setting are summarized in Table 2:

---

> ### Author Response · Authors · 2025-11-28
> **Author Responses to Reviewer szA3 (Part 2/3)**
>
> **Table 2. Pure IS Mode of SmartSAM on IS Benchmarks.**
>
> | Method                                   | DAVIS  | Amb-Occ |
> |------------------------------------------|:------:|:-------:|
> | Matcher-ViT-H (w/ visual reference, DINOv2-L) | 46.41 | 46.75  |
> | SAM-ViT-H                                | 45.97 | 34.35   |
> | **SmartSAM (no reference)**              | 59.71 | 43.94   |
> | SmartSAM (w/ visual reference, DINOv2-B) | 58.57 | **47.19** |
> | SmartSAM (w/ visual reference, DINOv2-L) | **70.74** | 46.64 |
>
> In the **no-reference** setting, we select the final mask as follows:
>
> 1. We **filter out** candidate masks that do not contain the user’s initial click.
> 2. Among the remaining candidates, we **choose the one with the highest IoU score predicted by SAM’s IoU head**.
>
> This procedure only re-ranks the masks produced by SAM and does **not** use any external exemplar.
>
> These results demonstrate that even **without any reference**, SmartSAM still significantly outperforms the base SAM model (+13.74 on DAVIS and +9.59 on Amb-Occ), showing that the improvements do **not** rely solely on additional exemplar inputs.
>
> ---
>
> **Q2 Limited technical complexity.**
>
> **A2.**  Thank you for raising this point.
>
> We agree that SmartSAM is intentionally **simple and training-free**, so that it can be **plugged on top of** any SAM-like model without retraining. However, the key idea behind our “inner interactions” is **uncertainty-guided and theoretically motivated**, rather than brute-force sampling.
>
> To understand why naive logits-based uncertainty is insufficient, we first measure how often the model’s **logit-level uncertainty** disagrees with **real-world ambiguity** on our datasets:
>
> | Dataset       | Error Rate (%) |
> |--------------|----------------|
> | DAVIS        | 29.18          |
> | PartImageNet | 26.53          |
> | Amb-Occ      | 34.18          |
> | **Overall**  | **30.77**      |
>
> As shown in Fig. 9 of the supplementary (and summarized in the table above), the logit distributions on ambiguous parts and normal regions are significantly different, revealing a clear mismatch between **logits-level uncertainty** and **true semantic ambiguity**. This motivates the need for **explicit, uncertainty-guided inner interactions** rather than randomly sampling clicks.
>
> Based on this observation, **Theorems 3.5 and 3.6** in the paper formally characterize how such uncertainty behaves and motivate our PIS and NCC policies: they **sample virtual clicks only within an uncertain band** (i.e., regions where the logits indicate neither clear foreground nor clear background) and adapt their behavior accordingly. Importantly, these theorems show that **even if some inner interactions are wrong (as reflected by the $\sim$30% error rate above), the agent can still recover a high-quality mask under mild assumptions**. This theoretical guarantee explains why a seemingly simple, training-free policy is sufficient to go beyond brute-force sampling in practice.
>
>
> ---
>
> **Q3. Limited novelty – the usage of text/visual reference is very limited.**
>
> **A3.**  Thank you for this insightful suggestion.
>
> We would like to clarify that **text/visual references are optional in SmartSAM and may not be available in many IS scenarios**, and this directly affects how they can be used.
>
> SmartSAM is designed as an **interactive segmentation (IS)** framework that is **not tied to image–mask exemplars**: it supports a **pure IS mode without any reference**, where only user clicks are available (see **(2)** in **A1**). For this reason, we deliberately keep the inner-click policies (PIS and NCC) **reference-agnostic**. They must behave identically with or without text/visual references; otherwise, the method would rely on information that does not exist in the no-reference IS setting, and the comparison to standard IS baselines would become unfair.
>
> We therefore **separate the roles** by design: PIS and NCC use only **uncertainty and geometry** to explore local variants around the user click, while text/visual references are **optionally** used **only in the final selection step** to resolve semantic ambiguity between candidate masks. This modular design keeps SmartSAM **training-free, fast, and directly pluggable** to any SAM-like model, while **allowing references to be completely optional** rather than mandatory inputs. In other words, the limited use of references is a **deliberate design choice to preserve a fair, reference-free IS mode**, not a restriction of the framework.

---

> ### Author Response · Authors · 2025-11-28
> **Author Responses to Reviewer szA3 (Part 3/3)**
>
> ---
>
> **Q4.** **Update the intro to better show the motivation of the work.**
>
> **A4.** Thank you for this helpful remark.
>
> First, we would like to reiterate that **SmartSAM is an interactive segmentation (IS) method**, and it supports a **pure IS mode without any text/visual reference**, where only user clicks are used (see A1 and A3). **Text/image references are therefore optional** in our framework, rather than mandatory inputs as in many FSS/personalized methods.
>
> Our intention in the introduction was not to suggest that **the user’s intent is unknown to the user**, but rather that with only a single click or gaze the intent is often **under-specified to the model**. In Fig. 1, for example, the user consistently intends “the whole lady”, but a single click on the torso is also compatible with other hypotheses such as “only the dress”. This under-specification from the model’s perspective is what we refer to as ambiguity.
>
> We agree that the current wording in the introduction does not clearly **convey this intended meaning** and **does not emphasize enough that references are optional**. In the revised version, we **reorganize the introduction** to:
>
> 1. explicitly state that **SmartSAM is an IS framework with a pure no-reference mode**, and
> 2. describe text/image references as an **optional additional channel** for expressing user intent, which SmartSAM uses only in the final mask-selection stage.
>
> For example, we revise the relevant sentence to:
>
> > "Therefore, we propose a training-free SmartSAM method to resolve ambiguity with a single click and one visual or textual reference, ..."
>
> **->**
>
> > “Therefore, we propose a training-free SmartSAM method to resolve ambiguity with a single click and one optional reference in either visual or textual form, ...”

---

### Official Review · Reviewer_buK9 · 2025-11-03

**Soundness:** 3
**Presentation:** 3
**Contribution:** 3
**Rating:** 6
**Confidence:** 2

**Summary:**

This paper proposes a novel interactive segmentation framework that utilizes the model's intrinsic knowledge to segment ambiguous objects effectively.

**Strengths:**

1 The question is interesting, and the idea is well organized.

**Weaknesses:**

1. More visualized comparison results with state-of-the-art models should be added.
2. Please also add a time comparison with state-of-the-art models
3. Please revise the writing of the methods part to make it easy to follow

**Questions:**

See the weakness

---

> ### Author Response · Authors · 2025-11-28
> **Author Responses to Reviewer buK9 (Part 1/2)**
>
> ---
>
> We would like to sincerely thank you for your efforts and valuable comments to improve our work!
>
> Below we address your concerns.
>
> ---
>
> **Q1. More visualized comparison results with state-of-the-art models should be added.**
>
> **A1.** Thank you for pointing this out.
>
> **Yes,** we will add more **visual results** comparing our method with other SOTA **Interactive Segmentation (IS), Personalized Segmentation (PS), Few-Shot Segmentation (FSS), and Open-Vocabulary Segmentation (OVS)** methods in **Fig. 18** of the revised supplementary materials. Some representative observations are:
>
> - **Compared with other baseline IS methods:** SAM-based IS methods often focus only on part of the target object, while SmartSAM is able to segment the entire object.
> - **Compared with PS/FSS methods:** compared with the PS method **PerSAM**, SmartSAM successfully segments the person riding the bike, whereas SAMWISE mistakenly highlights the black bag.
> - **Compared with OVS methods:** compared with the OVS method **SAMWISE**, SmartSAM correctly segments the person riding the horse, while SAMWISE fails to locate the correct person since it does not support human clicks as prompts.
>
> ---
>
> **Q2. Please also add a time comparison with state-of-the-art models.**
>
> **A2.** Thank you for this helpful remark.
>
> **Yes,** we have **added time-comparison experiments** using two metrics: (i) **time to reach 90% IoU (seconds@90)** and (ii) **time cost of the first click**. The **main result** is that SmartSAM **reduces the time needed to obtain high-quality masks**, while introducing only about **+0.04 s** additional time for the first click on average. The detailed results are as follows.
>
> The first table shows that SmartSAM **saves more than 1 second per image on average**:
>
> **Table 1. Time Cost of Baseline Models and SmartSAM Obtaining 90-IoU-Quality Masks.**
>
> | Baseline | DAVIS  | PartImageNet  | Amb-Occ  | Average Save Per Image (s) |
> |----------|--------|---------------|----------|----------------------------|
> | SAM      | 8.60   | 9.21          | 19.10    | -                          |
> | **w/ SmartSAM** | 8.34   | 8.22          | 15.93    | 1.47                       |
> | FocSAM   | 7.94   | 7.46          | 11.90    | -                          |
> | **w/ SmartSAM** | 7.86   | 6.77          | 10.97    | 0.57                       |
> | HQSAM    | 7.65   | 8.46          | 18.21    | -                          |
> | **w/ SmartSAM** | 7.29   | 8.10          | 15.45    | 1.16                       |
>
> > **Note.** The average human reaction time for a single human–computer click is **1.5 seconds** [1,2].
>
> The second table shows that, compared with the SOTA FSS method **Matcher**, SmartSAM is noticeably faster. Compared with other SOTA IS methods, SmartSAM requires only about **0.04 seconds** more on average for the first click.
>
> **Table 2. Time Cost of the 1st Click on an RTX 4090 GPU with Batch Size 2.**
>
> | Method                         | Backbone | DINO  | Time (s) |
> |--------------------------------|----------|-------|----------|
> | Matcher [1]                    | ViT-H    | ViT-L | 1.51     |
> | HRSAM                          | ViT-H    | -     | 0.91     |
> | HQSAM                          | ViT-H    | -     | 0.90     |
> | SAM                            | ViT-H    | -     | 0.91     |
> | **w/ SmartSAM** (Parallel $\dagger$)   | ViT-H    | ViT-B | 0.95     |
> | **w/ SmartSAM** (Sequential $*$)       | ViT-H    | ViT-B | 0.93     |
> > $\dagger$: get the SAM feature of the target image and Dino features of both target and reference parallelly.
>
> > $*$: first get the SAM features and then the Dino features.

---

> > ### Author Response · Authors · 2025-11-28
> > **Author Responses to Reviewer buK9 (Part 2/2)**
> >
> > ---
> >
> > **Q3.** **Please revise methodology part.**
> >
> > **A3**
> > Thank you for this suggestion.
> >
> > We agree that the current methodology section **needs more detailed descriptions**, and we have **substantially revised it**. The revisions mainly focus on:
> > - providing a **more accurate description of how SmartSAM works** in the main paper;
> > - adding **more implementation details** in the revised supplementary materials; and
> > - including **additional visual figures** that illustrate how SmartSAM generates the “inner clicks”.
> >
> > Below we summarize the key components of SmartSAM.
> >
> > **PIS (Priori Initial Sampler).**
> > - We first sample `num_points` candidate clicks around the user’s click according to Eq. (1) in the main paper.
> > - We then sample `num_boxes` candidate boxes using the same distribution, where the box size is drawn from a predefined scale distribution and the aspect ratio from another distribution (see line 207 of the revised main paper.).
> > - For each box $(x_1, y_1, x_2, y_2)$, we expand it if necessary to ensure that the user’s click $(x, y)$ is always included inside the box.
> >
> > **NCC (Next Chain Clicker).**
> > - NCC only issues clicks within **valid regions**, defined as follows:
> >   - For a **positive** click, the valid region is the current predicted mask.
> >   - For a **negative** click, the valid region is outside the mask but inside a rectangle that is $1.5\times$ larger than the bounding box covering the mask (this factor is a hyperparameter).
> > - Given the IoU score $h_{\text{iou}}$ predicted by SAM’s `iou_head`, NCC decides whether to issue a positive (label 0) or negative (label 1) click by sampling from a Bernoulli distribution $Bernoulli(h_{\text{iou}})$. Intuitively:
> >   - When SAM predicts a **low** score, the current mask may only cover part of the true object, so additional **positive** clicks are needed to add missing regions.
> >   - When SAM predicts a **high** score, the mask is already of high quality and roughly covers the whole object, so both **add-on** and **cut-off** operations are reasonable.
> >
> > ---
> >
> > **References**
> >
> > [1] A. Bearman, O. Russakovsky, V. Ferrari, and L. Fei-Fei, "What’s the point: Semantic segmentation with point supervision," in *ECCV*, 2016.
> >
> > [2] T. F. Chan and L. A. Vese, "Active contours without edges," *IEEE TIP*, 2001.

---

### Author Response · Authors · 2025-12-02
**SmartSAM — Rebuttal Summary for AC (Part 1/2)**

# **SmartSAM — Rebuttal Summary**

We thank all reviewers for their constructive comments. We have substantially revised the main paper and the supplementary material, and added new experiments, analyses, and clarifications. Below we summarize how we addressed the core concerns from each reviewer.

---

# **Reviewer-wise Summary of Concerns and Resolutions**

The reviewer **R1 `buK9` (Rating: 6)** mainly asks for stronger qualitative evidence and explicit time-cost analysis, as well as clearer exposition of the method. We address these concerns by adding extensive visual comparisons across IS/PS/FSS/OVS baselines, introducing seconds@90 and first-click latency (showing >1 s saving per image with only +0.04 s overhead), and rewriting the PIS/NCC/IoU-head parts with a concise pipeline diagram to improve clarity.

The reviewer **R2 `szA3` (Rating: 4)** is confused about the task setting (treating SmartSAM as FSS), notes the absence of COCO-20i and PASCAL-5i results, and questions the novelty and role of references. We clarify that SmartSAM is an IS agent with a strong reference-free mode, add COCO-20i and PASCAL-5i (1-shot, fold-0) experiments showing 4–5 mIoU gains over SAM, and emphasize that our main contribution is an ambiguity-resolution agent (PIS+NCC+IoU-head) rather than simple reference similarity.

The reviewer **R3 `KASf` (Rating: 4)** assumes that SmartSAM relies on exemplar references, finds our DAVIS numbers suspiciously low, and is not fully convinced about the novelty of the “agent”. We clarify that SmartSAM already brings large gains over SAM in pure IS mode without any references, explain that our DAVIS results follow the stricter FocSAM merged-foreground protocol, and provide new ablations, theory, timing and visual comparisons to demonstrate that the ambiguity-resolution agent is both effective and genuinely beyond a rule-based wrapper.

The reviewer **R4 `EcQ9` (Rating: 2, promises to update the score once misunderstandings are clarified)** finds the idea interesting but is concerned about robustness to perturbed / mislocated clicks, the practicality of annotation workflows, and how inner clicks are generated in realistic scenarios. We address these points by adding robustness experiments (TETRIS-style attacks, Amb-Occ natural deviations, and stronger perturbations on PartImageNet), clarifying recommended workflows for few-class vs many-class settings (reference mode vs pure IS), and providing additional explanation and visualizations of the PIS+NCC inner-click behavior, after which the reviewer indicates they will reconsider their score.

---

## **Reviewer R1 (`buK9`)**

1. **Visual comparisons.**
   We added extensive visual comparisons across IS / PS / FSS / OVS baselines (SAM-based IS, PerSAM, Matcher, SAMWISE, etc.), highlighting cases with ambiguity (occlusions, multiple instances, clutter) where SmartSAM reduces the need for extra clicks.

2. **Time-cost analysis.**
   We introduced **seconds@90** (time to reach 90% IoU) and **first-click latency**. SmartSAM saves **>1 second per image** on average at 90% IoU, while adding only **+0.04 seconds** for the first click and remaining **faster than Matcher**.

3. **Method clarity.**
   We rewrote the descriptions of PIS, NCC, and IoU-head filtering and added a concise diagram of the inner-click process. This makes the agent pipeline easier to follow and resolves the readability issues raised by R1.

---

## **Reviewer R2 (`szA3`)**

1. **References are optional, not required.**
   We clarified that SmartSAM has a strong **pure IS mode** that uses only user clicks and the SAM IoU head. In this mode, SmartSAM improves SAM by **+13.74 IoU on DAVIS** and **+9.59 IoU on Amb-Occ**, showing that the method does **not** rely on exemplar masks or texts. The reference mode only affects the **final selection** step.

2. **COCO-20i & PASCAL-5i results.**
   We added **COCO-20i and PASCAL-5i (1-shot, fold-0)** experiments with ProSAM, VRP-SAM, Matcher, SAM, and SmartSAM (with / without references). SmartSAM improves SAM by about **4–5 mIoU** on these FSS benchmarks, while we clearly state that SmartSAM is fundamentally an IS agent and FSS benchmarks are included for positioning.

3. **Novelty: ambiguity-resolution agent.**
   We emphasized that the novelty lies in modeling ambiguity resolution via an **agent (PIS + NCC + IoU-head)**, rather than just applying similarity. New ablations and theorems (e.g., recovery even when ~30% inner clicks are noisy) show that this design goes beyond simple rule-based heuristics.

---

> ### Author Response · Authors · 2025-12-02
> **SmartSAM — Rebuttal Summary for AC (Part 2/2)**
>
> ---
>
> ## **Reviewer R3 (`KASf`)**
>
> 1. **IS vs FSS setting.**
>    We clarified that SmartSAM is an **interactive segmentation** method, not an FSS model. COCO-20i / PASCAL-5i are used only to relate SmartSAM to FSS methods; our main focus remains IS benchmarks such as DAVIS, Amb-Occ, and PartImageNet.
>
> 2. **DAVIS protocol.**
>    We explained that we follow the **FocSAM DAVIS protocol**, where multiple objects are merged into a single foreground target, making the task strictly harder than standard DAVIS 2017-val and naturally giving lower scores. We added explicit description and figures to avoid further confusion.
>
> 3. **Fairness and novelty.**
>    We highlighted the strong **reference-free** gains (pure IS mode) to address fairness concerns, and pointed to the updated ablations, timing results, and qualitative comparisons. These show that the ambiguity-resolution agent provides consistent accuracy and robustness gains with small latency overhead, beyond a simple “rule wrapper” around SAM.
>
> ---
>
> ## **Reviewer R4 (`EcQ9`)**
>
> 1. **Robustness to perturbed clicks.**
>    We added robustness experiments under (i) **TETRIS-style attacked clicks** ($\sigma$ = 0.15), (ii) **Amb-Occ** with natural deviation, and (iii) **PartImageNet** with stronger perturbations. In all cases, SmartSAM is more robust than SAM and SAM2 and maintains clear gains under mislocated first clicks.
>
> 2. **Practical annotation workflows.**
>    We clarified two regimes: in **few-class settings** (10–30 classes), annotators can maintain a small exemplar pool and use the optional reference mode for faster disambiguation; in **many-class / open-vocabulary settings**, they can simply run SmartSAM in **pure IS mode**, where the agent still reduces the number of corrective clicks without any reference overhead.
>
> 3. **Inner-click behavior.**
>    We added explanation and visuals for how PIS and NCC operate when the first click is off-center, and noted that combining SmartSAM with human-click logs or clickability models (TETRIS/RClicks) is an interesting direction for future work. After these clarifications, R4 explicitly indicates willingness to reconsider their score.

---

### Meta-Review · Area_Chair_Qkcv · 2026-01-03

**Summary:**

Across reviewers, the main concerns focus on limited novelty, incomplete experimental validation, and unclear practical benefits.

a) Several reviewers note that the core idea — generating multiple segmentation hypotheses and selecting the best one using text or visual references — appears conceptually simple and largely incremental relative to existing reference-based and personalized segmentation methods.

b) The experimental evaluation is considered insufficient, as key standard benchmarks (e.g., COCO-20i and PASCAL-5i) and stronger state-of-the-art baselines are missing, making it difficult to assess comparative performance. Reviewers also question the fairness of comparisons, since SmartSAM benefits from additional information (reference images, masks, or more candidate hypotheses) relative to baselines such as SAM and other interactive or few-shot methods.

c) Practical applicability is another recurring concern: the reliance on external references, the absence of human user studies, unrealistic click simulation strategies, and missing runtime or time-efficiency comparisons limit confidence in real-world annotation speedups.

Minor concerns are raised c.f. method descriptions, requesting more visualized qualitative comparisons, and additional analysis to better demonstrate the strengths and limitations of the proposed approach.

**Reviewer Concerns:**

The authors' rebuttal addresses several reviewer concerns regarding experimental sufficiency and fairness by providing additional visualization results and quantitative evaluations across multiple dimensions: (a) computational efficiency and time costs; (b) evaluations on FSS benchmarks (COCO-20i and FSS-5i); (c) a no-reference mode for the proposed smartSAM, demonstrating the validity of the rule-based approach without relying on external exemplars; and (d) simulated "attacked first click" scenarios (mimicking real-world human-like interactions), where smartSAM outperforms the SAM baseline.

However, the rebuttal falls short in convincingly addressing the major concerns about technical contributions and novelty. I concur with the reviewers that the proposed method relies on straightforward rule-based mechanisms and similarity-based feature filtering, both of which have been extensively explored in the literature. In particular, as noted by reviewer EcQK regarding the practical significance of this work, sourcing references and exemplars could prove more tedious and time-consuming for users than simply adding a few extra clicks — a point the authors do not adequately rebut.

**Reviewer Scores:**

(a) Reviewer buK9 would likely maintain their original score, as the reviewer does not actively involve during the rebuttal period.
(b) Reviewer szA3 would maintain their original score, given that their primary concerns centered on technical novelty and contributions, which were only partially addressed by the authors and remain insufficiently rebutted.
(c) Reviewer KASf would maintain their original score for reasons similar to those of Reviewer szA3, with the rebuttal failing to fully alleviate doubts about the work's innovative aspects.
(d) Reviewer EcQK would either maintain their original score or potentially raise it to a 4, since their concerns regarding practical value and experimental validation have been partially addressed through additional evaluations and demonstrations, though key points like user effort in sourcing exemplars persist.

---

### Decision · Program_Chairs · 2026-01-26

Reject